# MRO: Enhancing Reasoning in Diffusion Language Models via Multi-Reward Optimization

**Chenglong Wang**[1]    **Yang Gan**[1][*]   **Hang Zhou**[1]    **Chi Hu**[2]
**Yongyu Mu**[1]    **Kai Song**[2]    **Murun Yang**[1]    **Bei Li**[1]    **Chunliang Zhang**[1,3]
**Tongran Liu**[4]    **Jingbo Zhu**[1,3]    **Zhengtao Yu**[5]    **Tong Xiao**[1,3][†]

[1]School of Computer Science and Engineering, Northeastern University, Shenyang, China
[2]ByteDance    [3]NiuTrans Research, Shenyang, China
[4]CAS Key Laboratory of Behavioral Science, Institute of Psychology, CAS, Beijing, China
[5]Kunming University of Science and Technology
{clwang1119, zzhu8250}@gmail.com    {xiaotong, zhujingbo}@mail.neu.edu.cn

## Abstract

Recent advances in diffusion language models (DLMs) have presented a promising alternative to traditional autoregressive large language models (LLMs). However, DLMs still lag behind LLMs in reasoning performance, especially as the number of denoising steps decreases. Our analysis reveals that this shortcoming arises primarily from the independent generation of masked tokens across denoising steps, which fails to capture the token correlation. In this paper, we define two types of token correlation: *intra-sequence correlation* and *inter-sequence correlation*, and demonstrate that enhancing these correlations improves reasoning performance. To this end, we propose a **M**ulti-**R**eward **O**ptimization (MRO) approach, which encourages DLMs to consider the token correlation during the denoising process. More specifically, our MRO approach leverages test-time scaling, rejection sampling, and reinforcement learning to directly optimize token correlation with multiple elaborate rewards. Furthermore, we introduce a step-wise group reward optimization approach to mitigate reward variance during the reward optimization. Through extensive experiments, we demonstrate that MRO not only improves reasoning performance but also achieves significant denoising speedups while maintaining high performance across reasoning tasks.

## 1 Introduction

Large language models (LLMs) have recently made remarkable strides, profoundly impacting the entire field of artificial intelligence. Almost all of these models share a typical recipe: learn a model that maximizes data likelihoods using an autoregressive (AR) paradigm [1, 2, 3]. This AR paradigm enables LLMs to perform exceptionally well in a wide range of downstream tasks, particularly demonstrating impressive abilities to solve complex reasoning problems using chain of thought (CoT) methods [4, 5, 6, 7, 8]. Models such as OpenAI's o1 [9], Qwen2.5 [10], and DeepSeek-R1 [11] are at the forefront of this progress. Despite its empirical success, the AR paradigm has inherent limitations. During decoding, AR models generate tokens in a left-to-right, token-by-token manner, which constrains both the efficiency and flexibility of generation [12, 13]. Furthermore, the generation process heavily depends on previously generated tokens, which often leads to error accumulation (also known as exposure bias) [14, 15].

---

[*]Work was done when Yang Gan was interning at ByteDance.
[†]Corresponding author.

39th Conference on Neural Information Processing Systems (NeurIPS 2025).

To overcome these limitations, recent research has explored alternative approaches to developing language models [16, 17, 18]. Among these, diffusion language models (DLMs) have emerged as a promising and competitive direction [13, 19, 20, 21]. Unlike traditional AR LLMs, DLMs generate sequences by iteratively predicting multiple masked tokens in parallel at each intermediate denoising step, enabling bidirectional and controllable generation while improving sampling efficiency. As part of ongoing efforts in this area, substantial progress has been made in pretraining and scaling DLMs, which have demonstrated strong capabilities in text generation tasks [22, 23]. More recently, Ye et al. [24] have proved the potential of DLMs in solving planning tasks, including Countdown and Sudoku.

Despite their potential, DLMs still underperform AR models by a large margin in reasoning tasks, as also evidenced in Table 1. As a result, developing DLMs with strong reasoning capabilities remains an open research problem. To address this, DoT [25] introduces the CoT technique into the denoise process of the DLM. Instead of denoising the entire sequence at each step, DoT distributes the reasoning process across denoising steps, allowing each step to correspond to a distinct reasoning stage. However, this approach still partially relies on the AR paradigm, making it susceptible to error accumulation during reasoning. Additionally, *d1* [26] improves the reasoning performance of DLMs by fine-tuning them with a policy gradient algorithm.

All of the existing efforts, however, overlook a fundamental factor contributing to the poor reasoning performance: the independent generation of multiple masked tokens across denoising steps fails to capture the dependencies among them in CoT-style reasoning. We argue that this independence introduces inconsistencies in the reasoning path, ultimately leading to incorrect outcomes, as the generated tokens may lack correlation with one another. Specifically, we define two types of token correlation in DLMs: *intra-sequence correlation* and *inter-sequence correlation*.

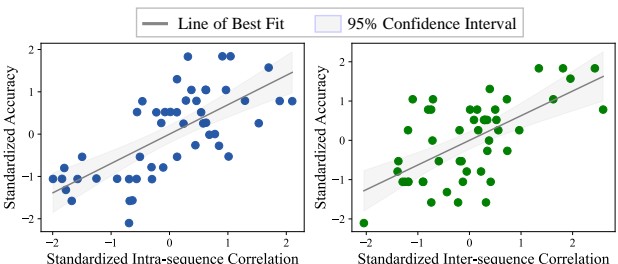

Figure 1: Visualization of the relationship between token correlation and reasoning accuracy. The results show that higher intra- or inter-sequence correlation tends to yield higher reasoning performance.

Intra-sequence correlation measures the degree of dependency among tokens generated within a single denoising step. In contrast, inter-sequence correlation captures the alignment between sequences generated at different denoising steps. Furthermore, we examine the relationship between the correlation and reasoning accuracy by performing 50 decoding runs on a subset of GSM8K, as shown in Figure 1. From the results, we observe that higher token correlation leads to more accurate reasoning outcomes. This finding aligns with the observation in Lightman et al., [27]'s work, that suggests that inconsistencies within the reasoning path often lead to incorrect outcomes.

*How can we equip DLMs with enhanced token correlation in reasoning?* One straightforward approach is to scale the training data and model size [22, 23], enabling DLMs to better capture the correlation between tokens. While this approach does not directly optimize the correlation for DLMs, scaling the data and model size could potentially enhance it by improving the accuracy of masked token predictions. Unfortunately, such scaling often comes with significantly increased requirements for labeled data and computational resources. In this paper, we propose an approach that enhances the token correlation without these burdens. Our approach, **M**ulti-**R**ewards **O**ptimization (MRO), enables DLMs to generate reasoning paths with an emphasis on token correlation.

To develop MRO, we begin by designing multiple rewards and incorporating test-time scaling, rejection sampling, and reinforcement learning to optimize these rewards simultaneously. Intuitively, the optimization objective during DLM training is to predict masked tokens, but the decoding joint distribution is parameterized as a product of token-wise independent distributions. To this end, MRO enhances the correlation through a multi-reward optimization mechanism, capturing dependencies among tokens and bridging this gap. Furthermore, we find that MRO encounters the issue of reward variance when handling multiple rewards simultaneously. To address this, we introduce a step-wise group reward optimization (namely SGRO) approach to optimizing DLMs with rewards. We provide a theoretical analysis of how our SGRO reduces the reward variance from a potential-based reward shaping perspective, as shown in Appendix B.

Through extensive experiments on MRO with test-time scaling, rejection sampling, and reinforcement learning, we show that multi-reward optimization is effective. For example, MRO yields an average improvement of 3 points across various reasoning tasks with test-time scaling. As another bonus, MRO enables DLMs to generate high-quality reasoning paths with fewer denoising steps, thereby accelerating the overall decoding process.

## 2 Related Work

### 2.1 Diffusion Language Models

Building on previous advancements in diffusion models for image generation [28, 29, 30], recent efforts have attempted to explore the application of diffusion models for text generation tasks. For example, text-based continuous diffusion models [31, 32] have introduced an embedding function to map discrete text into a continuous space. Furthermore, to accommodate the inherently discrete nature of text, researchers have explored discrete diffusion models for language modeling [33]. Despite their significant success, challenges remain, particularly with scalability. More recently, the masked diffusion model [21, 22] has emerged as a breakthrough instance of the discrete diffusion model. For example, DiffuLLaMA [34] extended the masked diffusion model by initializing it with parameters from LLaMA models. In another line of research, based on these established DLMs, researchers attempted to optimize the denoising process to further enhance them. This includes integrating the CoT approach [25] and AR paradigm [18] during the denoising process. Excitingly, in the context of scaling DLMs, LLaDA [23] and Dream [24] showed that it is possible to scale DLMs to the 8B and 7B sizes, respectively, with pre-training and SFT training, achieving text generation capabilities comparable to AR LLMs. Motivated by the recent success of DeepSeek-R1 [11], which leverages reinforcement learning to scale the reasoning capability, Zhao et al. [26] proved the effectiveness of applying reinforcement learning to DLMs for improving reasoning performance. Although previous work improves the performance of DLMs on various text generation tasks, they often overlook a fundamental limitation: the inherent parallel and independent generation of tokens in DLMs, which leads to weak token correlations across denoising steps. This limitation can negatively impact tasks that require high consistency, such as CoT-style reasoning, where coherence and alignment of content are critical. Researchers have been aware of this [35], but it is still rare to see studies on this issue.

### 2.2 Reward Optimization for Language Models

Reward optimization is a fundamental concept in reinforcement learning that helps agents make better decisions. In the context of LLMs, reward optimization can guide the generated content to better align with human preferences. A widely used approach for reward optimization is reinforcement learning from human feedback [5, 36, 37, 38], where a reward model is trained based on human feedback, and the LLM is optimized relative to this reward using algorithms like proximal policy optimization [39] and rejection sampling [40]. This area has seen significant progress, with recent examples including models developed by OpenAI [9] and DeepSeek [11], which employ reward optimization approaches to teach models to human-like thinking and complex reasoning. Building on this foundation, there is a growing body of work exploring how to design reward functions that enable LLMs to perform more effectively across a wide range of tasks [41, 42].

## 3 Preliminaries

In this section, we outline some basic concepts and notation of diffusion language models.

### 3.1 Training Diffusion Language Models

Discrete diffusion models [33] have emerged as a promising approach for language modeling, providing an alternative to traditional AR models. One of the most popular methods in this context is the "masked" approach, also known as masked diffusion models [21, 43, 22]. More excitingly, [23] scales masked diffusion models to a 7B parameter model and integrates unsupervised pre-training with instruction fine-tuning during the training of DLMs, achieving an impressive text generation performance. Specifically, at the pre-training stage, the training objective involves minimizing a cross-entropy loss computed only for masked tokens. Given an unlabeled training sequence $x_0$, a

time step $t$ with a masking ratio $o_t$ sampled uniformly from $[0, 1]$, and the sequence $x_t$ is obtained by masking each token independently with probability $t$. The loss function is

$$\mathcal{L}_{\text{pre}}(\theta) = -\mathbb{E}_{o_t, x_0, x_t} \left[ \frac{1}{o_t} \sum_{i=1}^{L} \mathbf{1}[x_t^i = M] \log \text{Pr}_\theta(x_0^i \mid x_t) \right] \tag{1}$$

where $L$ is the length of the sequence $x_0$, $x_0^i$ is the $i$-th token, $M$ is the masked token, and $\mathbf{1}[\cdot]$ is the indicator function. Here, $\text{Pr}_\theta(\cdot)$ denotes the mask predictor, and $\theta$ is the set of its parameters. It is typically parameterized by a stacked Transformer model without a causal mask. This loss function encourages the model to accurately predict the original tokens from the masked sequence. The basic idea for pre-training DLMs is similar to that of pre-training AR LLMs. Both aim to capture linguistic knowledge from large amounts of unsupervised data, whereas AR LLMs rely on next-token prediction, and DLMs use masked token prediction, a strategy proven effective in encoder-only language models, such as BERT [44] and RoBERTa [45].

After pre-training, we can employ labeled data to enhance DLMs' ability to follow instructions with supervised fine-tuning (SFT). This phase involves training the model on labeled paired data $(p_0, r_0)$, where $p_0$ is the prompt, such as "*If $x + 5 = 12$, what is the value of $x$?*" and $r_0$ is the corresponding response, such as "*The answer is 7*". During SFT, the prompt $p_0$ is kept unchanged, and tokens in the response $r_0$ are masked independently. The masked response $r_t$ is then fed into the pre-trained mask predictor to compute the loss:

$$\mathcal{L}_{\text{sft}}(\theta) = -\mathbb{E}_{o_t, p_0, r_0, r_t} \left[ \frac{1}{o_t} \sum_{i=1}^{L_r} \mathbf{1}[r_t^i = M] \log \text{Pr}_\theta(r_0^i \mid p_0, r_t) \right] \tag{2}$$

where $L_r$ is the length of $r_0$. The training stage is similar to the pre-training, with the key difference being that the model now learns to predict the response tokens conditioned on the given prompt.

## 3.2 Applying Diffusion Language Models

A fundamental application of DLMs is cloze (fill-in-the-blank) tasks, which align with the training objective. However, since the ultimate goal is to replace AR LLMs, a key expectation is that DLMs should also be capable of generating textual sequences. To achieve this, we typically use masked token prediction to simulate the denoising step. More specifically, given an unseen prompt $p_0$, the model starts with a fully masked response $r_T$ and iteratively predicts the masked tokens to fully reconstruct the response $r_0$ over $T$ time steps. The process also involves re-masking a fraction of the predicted tokens at each step to ensure that the reverse process aligns with the forward process.

## 4 Multi-Reward Optimization for Diffusion Language Models

### 4.1 Problem Definition

During the DLM decoding process, we use the learned denoising distribution $\text{Pr}_\theta(\cdot)$ to construct the decoding distribution $Q_\theta([r_{T-1}, \cdots, r_0] \mid p_0, r_T)$, where $[r_{T-1}, \ldots, r_0]$ denotes the sequence of intermediate responses progressively refined through the iterative denoising steps. This process predicts the masked tokens across $T$ denoising iterations and can be expressed as

$$Q_\theta([r_{T-1}, \cdots, r_0] \mid p_0, r_T) = \underbrace{\prod_{t=T}^{1} U_\theta(r_{t-1} \mid p_0, r_t)}_{\text{inter-sequence correlation}} = \prod_{t=T}^{1} \underbrace{\prod_{i=1}^{L_r} \mathbf{1}[r_{t-1}^i = M] \text{Pr}_\theta(r_{t-1}^i \mid p_0, r_t)}_{\text{intra-sequence correlation}} \tag{3}$$

where $U_\theta(\cdot)$ denotes the denoising distribution that jointly predicts all masked tokens at step $t - 1$ conditioned on the current partially denoised sequence $r_t$ and the initial prompt $p_0$. While this approach effectively constructs the decoding distribution $Q_\theta(\cdot)$ through iterative denoising, it fails to capture the dependencies between tokens across denoising steps. Consequently, $\text{Pr}_\theta(\cdot)$ cannot perfectly approximate the true decoding distribution, leading to weakened token-level correlations and inconsistencies within the generated sequence. These inconsistencies can be particularly harmful in reasoning-intensive tasks (e.g., CoT-style reasoning) where precision and coherence are essential. As shown in Eq. 3, we define these correlations as intra-sequence correlation and inter-sequence correlation, with their definitions outlined below:

- *Intra-sequence Correlation.* This refers to the dependencies among tokens generated within a single denoising step. In reasoning tasks, it measures how well these tokens collaborate to form a coherent and logical segment of the reasoning path, ensuring fluency and consistency.
- *Inter-sequence Correlation.* It captures the alignment between tokens generated at different denoising steps. This correlation ensures that the sequence generated at time step $t - 1$ is consistent with the sequence generated at time step $t$, leading to a coherent reasoning path.

## 4.2 Multi-Reward Optimization

We aim to design multiple rewards that optimize token correlations during the DLM decoding process. We begin by describing separate reward designs targeting intra-sequence and inter-sequence correlations, respectively, and then explain how to combine these rewards in the optimization process.

### 4.2.1 Intra-sequence Correlation Rewards

For token correlation within a single sequence at a denoising step, we evaluate it from two aspects. First, during decoding, we assess whether the tokens predicted in parallel at one denoising step exhibit strong correlations. Second, based on these generated tokens, we check whether the current sequence forms a coherent and readable response.

**Token Verification Reward.** To address the first aspect, we design a token verification reward (denoted as $R^{\text{tv}}$). This basic idea is that, at each denoising step, after predicting the masked tokens, we re-enter the model to verify the correlations between the generated tokens. Specifically, consider an instance with $N$ masked tokens: at the $t$-th denoising step, given the prompt $p_0$ and the intermediate response $r_t$, the DLM predicts the masked tokens $\{r_{t-1}^{m_1}, r_{t-1}^{m_2}, \ldots, r_{t-1}^{m_N}\}$, which together form $r_{t-1}$, where $\{m_1, m_2, \ldots, m_N\}$ represents the indices of the masked token positions. We then use the token verification method to compute the reward $R^{\text{tv}}$ for this time step, denoted as $R_t^{\text{tv}}$:

$$R_t^{\text{tv}} \quad = \quad \frac{1}{N} \sum_{n=1}^{N} \text{Pr}_\theta(r_t^{m_n} \mid p_0, r_{t-1}/r_{t-1}^{m_n}) \tag{4}$$

where $r_{t-1}/r_{t-1}^{m_n}$ denotes the response where only the token $r_{t-1}^{m_n}$ is masked from $r_{t-1}$. Furthermore, we provide a theoretical proof that the token verification reward enhances intra-sequence token correlations from the perspective of mutual information (see Property 3 in Appendix B). In practice, however, calculating this reward introduces additional computational overhead. To mitigate this issue, we adopt three optimization strategies as follows. First, for each denoising step, we randomly sample a subset of tokens for verification instead of checking all tokens. Second, we leverage GPU parallelism to compute the probabilities of different masked tokens simultaneously, thereby reducing time cost. Third, we introduce a step-wise group optimization strategy to further lower the reward computation overhead, as described in Section 4.3.

**Perplexity Reward.** For the second aspect, considering perplexity is widely used as a measure of textual sequence consistency and readability [46], we design a perplexity reward (denoted as $R^{\text{ppl}}$). Specifically, at the $t$-th denoising step, we use a pre-trained AR language model to compute the perplexity of the response $r_{t-1}$ generated at that step, and then compute $R_t^{\text{ppl}}$ as follows

$$R_t^{\text{ppl}} \quad = \quad \frac{\max\{C_{\text{ppl}} - \text{PPL}(r_{t-1}), 0\}}{F_{\text{ppl}}} \tag{5}$$

where $\text{PPL}(\cdot)$ denotes the perplexity computation function, $C_{\text{ppl}}$ is a fixed upper bound constant controlling the maximum reward value, and $F_{\text{ppl}}$ is a scaling factor used to ensure that the range of $R_t^{\text{ppl}}$ is comparable to that of other rewards. Here, we use $\texttt{lmppl}^3$ to implement it.

### 4.2.2 Inter-sequence Correlation Rewards

For token correlation across different sequences during the denoising steps, we primarily assess whether the combination of these steps results in a high-quality response. Specifically, for reasoning

---

[3]`https://github.com/asahi417/lmppl`

tasks, we build on recent work [11] to evaluate the quality of CoT-style reasoning generated by DLMs, focusing on both CoT format and accuracy. Given a prompt $p_0$, once we obtain $r_0$, the corresponding reward $R_0^{\mathrm{q}}$ is computed as follows

$$
R_0^{\mathrm{q}} = \begin{cases} 2, & \text{if the } r_0 \text{ follows the required format and its answer is correct} \\ 1, & \text{if the } r_0 \text{ follows the required format but its answer is incorrect} \\ 0, & \text{if the } r_0 \text{ does not follow the required format} \end{cases} \tag{6}
$$

Following [11], we define a format reward, which checks whether the CoT strictly appears within the "<think> </think>" tag and whether the answer strictly appears within the "<answer> </answer>" tag. Note that our inter- and intra-sequence correlation rewards are not entirely independent. A stronger intra-sequence correlation reward, which enforces local token-level coherence, can indirectly enhance the inter-sequence reward by improving the overall fluency and consistency of each intermediate response. Nevertheless, the two rewards focus on different aspects of correlation: the intra-sequence reward emphasizes token-level dependencies within a single response, while the inter-sequence reward evaluates the global consistency and task-level quality across denoising steps.

### 4.2.3 Reward Optimization via Markov Decision Process

To achieve the reward optimization, we model the iterative denoising process of DLMs as a Markov Decision Process (MDP) [47]. In this formulation, each denoising step corresponds to an action, and the goal is to optimize the rewards at each step by considering both intra-sequence and inter-sequence token correlations. Formally, we define the key MDP components as follows:

$$
s_t \triangleq (p_0, r_t, t), \quad a_t \triangleq r_{t-1}, \quad \pi_\theta(a_t \mid s_t) \triangleq U_\theta(r_{t-1} \mid p_0, r_t) \tag{7}
$$

where $s_t$ is the state at the $t$-th step, which includes the current time step $t$, the given prompt $p_0$, and the generated tokens at step $r_t$. $a_t$ represents the action taken at step $t$, which corresponds to the predicted response $r_{t-1}$. $\pi_\theta(a_t \mid s_t)$ denotes the policy of the DLM, which predicts the tokens at the previous time step $t-1$ based on the current tokens $r_t$ and the prompt $p_0$. Based on this MDP formulation, we can easily apply optimization algorithms like policy gradient [48] to maximize an arbitrary reward (denoted as $R(s_t, a_t)$) during the denoising process.

Here, we construct $R(s_t, a_t)$ by incorporating intra-sequence and inter-sequence correlation rewards. Specifically, we define the intra-sequence correlation rewards in a potential-based reshaping reward [49]. At each denoising step $t$, after generating the tokens $r_{t-1}$, we shape the reward by considering both the token verification reward $R^{\mathrm{tv}}$ and the perplexity reward $R^{\mathrm{ppl}}$. Additionally, we treat the inter-sequence correlation reward $R^{\mathrm{q}}$ as a delayed reward, as the alignment between sequences generated across different denoising steps becomes evident only after the entire sequence is generated. Based on this reward allocation scheme, we can define the $R(s_t, a_t)$ by

$$
R(s_t, a_t) \triangleq \begin{cases} R_t^{\mathrm{q}} + R_t^{\mathrm{tv}} + R_t^{\mathrm{ppl}}, & \text{if } t = 0 \\ R_t^{\mathrm{tv}} + R_t^{\mathrm{ppl}}, & \text{otherwise} \end{cases} \tag{8}
$$

The benefit of this formulation is that if we use a standard sampling strategy with $U_\theta(r_{t-1} \mid p_0, r_t)$ as described in Eq. 3, this policy $\pi$ can optimize this reward and adjust the parameter $\theta$ when using algorithms like policy gradient. This enables DLMs to directly optimize token correlations, which are typically absent during pre-training and SFT training processes.

### 4.3 Step-wise Group Reward Optimization

As mentioned earlier, we combine the rewards using a potential-based reshaping approach. Let $\Phi(\cdot)$ denote the potential function. We can express the difference as $\lambda\Phi(s_{t+1}) - \Phi(s_t) = R_t^{\mathrm{q}} + R_t^{\mathrm{tv}}$, where $\lambda$ is the discount factor. Thus, we have $R(s_t, a_t) = \hat{R}(s_t, a_t) + \lambda\Phi(s_{t+1}) - \Phi(s_t)$, where $\hat{R}(s_t, a_t)$ denotes the original sparse reward. While potential-based reshaping can effectively optimize token correlations, [50] shows that, in the case of longer decision trajectories in MDPs, this approach can lead to higher reward variance during the optimization of parameters. We establish a similar result for our multi-reward optimization in Property 1.

**Property 1.** *Under potential-based reward shaping, the expected reward remains the same, i.e.,* $\mathbb{E}[R(s, a)] = \mathbb{E}[\hat{R}(s, a)]$, *but* $\mathrm{Var}(R(s, a))$ *is higher than* $\mathrm{Var}(\hat{R}(s, a))$.

The proof can be found in Appendix B. This conclusion highlights a limitation in our reward combination method. To address this limitation, we propose the Step-wise Group Reward Optimization (SGRO) approach. This approach groups $w$ denoising steps together, with each group providing a reshaped reward, thereby replacing the computation of the intra-sequence correlation reward at each individual denoising step. By reducing the number of reshaped reward computations, SGRO helps lower reward variance. Specifically, we further prove that $\text{Var}(R(s,a)) > \text{Var}(R^{(w)}(s,a))$, where $R^{(w)}(\cdot)$ denotes the grouped reward after applying SGRO (see Property 2 in Appendix B). In addition to variance reduction, we find that SGRO also decreases the frequency of reward evaluations, thus reducing the computational overhead of reward estimation, as described in Appendix D.

## 5 Experiments

We evaluate the effectiveness of our multi-reward optimization (MRO) approach with various optimization algorithms, including test-time scaling, rejection sampling, and reinforcement learning.

### 5.1 Setups

We conducted our experiments using the LLaDA-8B-Instruct model [23], a state-of-the-art open-source DLM that employs a fully masked-based training approach for both pre-training and SFT training. Additionally, we performed experiments on the LLaDA-8B-Instruct-s1 model, a reasoning DLM derived by fine-tuning the pre-trained LLaDA-8B model with the s1 training dataset [51]. For evaluation, we considered five reasoning benchmarks across three categories: (1) *Mathematical reasoning*: GSM8K and MATH500; (2) *Scientific reasoning*: GPQA, which focuses on biology, physics, and chemistry reasoning; (3) *Logical reasoning*: 4×4 Sudoku and the Countdown task with 3 numbers. More experimental details can be found in Appendix C.

### 5.2 Baselines

We compared our MRO with several strong baselines: the *LLaDA-8B-Instruct* and *LLaDA-8B-Instruct-s1* models, which are two DLMs that have not undergone reward optimization (denoted as LLaDA and LLaDA-s1) and *AR LLMs*, including serval open-source AR LLMs with around 8B parameters, such as LLaMA-3-8B-Instruction [52], Mistral-7B-Instruct [53], Deepseek-LLM-7b-Chat [11], and Qwen2.5-7B-Instruct [10].

### 5.3 Test-time Scaling

In this subsection, we examine whether our MRO enables DLMs to generate accurate reasoning outcomes. To achieve this, we perform test-time scaling to evaluate the effectiveness of MRO.

**Task Setup.** As illustrated in Figure 2, we performed beam search to explore better decoding results. At the $t$-th denoising step, we generated $k$ different responses $\{r_{t,1}, r_{t,2}, \cdots, r_{t,k}\}$ using temperature-based sampling, where $k$ was set to 4. We then computed the reward for each response using Eq. 8 and selected the response with the highest reward as the final result for that step. Note that in the test-time scaling experiment, we did not have access to the ground outcome when computing $R_0^{\text{q}}$. Therefore, we employed a majority voting approach to obtain a pseudo-ground outcome, which was then used to compute $R_0^{\text{q}}$. The temperature was set to 0.25, and we present the performance with other temperature values in Figure 5. We also tested different response lengths, including 64, 128, 256, and 512. For each setting,

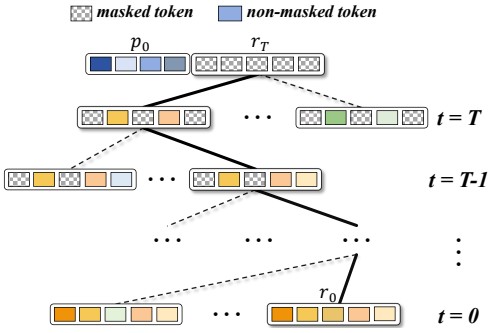

Figure 2: Illustration of the test-time scaling procedure in DLMs. The bolded line denotes that the response obtains the highest reward and is selected as the final output.

we set the number of denoising steps to half the generated response length. Additionally, we applied our SGRO, setting $w$ to 32, meaning that beam search with MRO was applied every 32 steps during

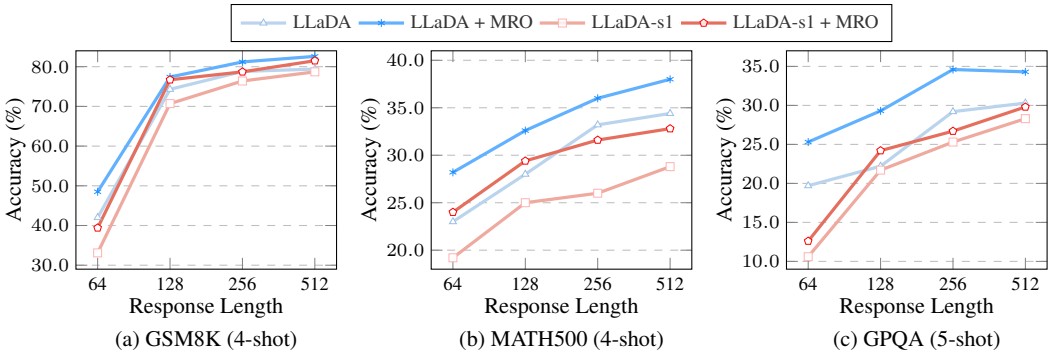

Figure 3: Accuracies (%) on three reasoning tasks (GSM8K, MATH500, and GPQA) with test-time scaling, showing results for different response lengths.

the beam search process. For evaluation, we adhered to the official evaluation procedure provided by LLaDA, which employs semi-autoregressive sampling by default [12]. The block lengths were set to 8, 64, and 64 for GSM8K, MATH500, and GPQA, respectively. The final accuracy metrics were obtained using the `lm-evaluation-harness`[4] toolkit.

**Results.**    As shown in Figure 3, MRO significantly improves reasoning performance across different response lengths and models. Specifically, when comparing models with and without MRO, we observe notable improvements in accuracy. Firstly, the token correlation rewards we introduced are proven to be effective. For example, in the GSM8K task, the LLaDA model with MRO achieves an accuracy of 82.6% at a response length of 512, whereas the LLaDA model without MRO reaches only 79.4%. Similarly, for the MATH500 task, LLaDA + MRO improves from 34.4% to 38.0% at a response length of 512. Secondly, the results also confirm the claim made in Section 4.1 that DLMs like LLaDA often miss important token correlations during decoding, which can limit their performance. These observations confirm that the proposed reward design effectively enhances the model's reasoning capability, providing a useful insight: *the designed reward could be leveraged to further optimize the DLM via parameter adjustment using techniques like reinforcement learning or rejection sampling*. We validate this insight in the following section.

## 5.4   Rejection Sampling

While test-time scaling effectively implements MRO, the decoding time is significantly scaled. Ideally, we aim to optimize the parameters of this DLM to consider the token correlation during the decoding stage. Here, we employ rejection sampling to achieve this goal.

**Task Setup.**    We used beam search and SGRO, as described in Section 5.3, to obtain a sampled sequence $[\hat{r}_T, \hat{r}_{T-1}, \ldots, \hat{r}_0]$ that yields high rewards. This sequence is then used for SFT training of our DLM. The corresponding loss function can be given by

$$\mathcal{L}_{\text{rs}}(\theta) \quad = \quad -\mathbb{E}_{o_t, p_0, \{\hat{r}_T, \hat{r}_{T-1}, \cdots, \hat{r}_0\}} \left[ \frac{1}{o_t} \sum_{i=1}^{L_{\hat{r}}} \mathbf{1}[\hat{r}_t^i = M] \log \Pr_\theta(\hat{r}_{t-1}^i \mid p_0, \hat{r}_t) \right] \tag{9}$$

where $L_{\hat{r}}$ denotes the length of the response $\hat{r}$. During our training process, we experimented with setting the beam size $k$ for each expansion step to 2 and 4. Furthermore, we found that computing the gradients for all denoising steps simultaneously consumed a large amount of GPU memory. To address this issue, we developed two strategies. First, we used gradient accumulation, where we computed the gradients for each individual step and accumulated them until the entire sequence was processed, after which backpropagation was performed. Second, instead of optimizing over the entire sequence, we selected a segment of denoising to optimize. The basic idea is that token correlations between different denoising steps can be captured not only by using the full denoising sequence, but also through combinations of smaller subsets.

---

[4]https://github.com/EleutherAI/lm-evaluation-harness

| Model/Length | GSM8K | | | MATH500 | | | GPQA | | | Countdown | | Sudoku |
|---|---|---|---|---|---|---|---|---|---|---|---|---|
| | 128 | 256 | 512 | 128 | 256 | 512 | 128 | 256 | 512 | 64 | 128 | 64 |
| LLaMA-3-8B-Instruction[†] | | 53.1 | | | 18.4 | | | 25.9 | | | 3.7 | 0.0 |
| Mistral-7B-Instruct[†] | | 36.2 | | | 13.1 | | | 24.7 | | | 1.2 | 0.0 |
| Deepseek-LLM-7b-Chat[†] | | 17.4 | | | 6.0 | | | 19.5 | | | 8.5 | 0.0 |
| Qwen2.5-7B-Instruct[†] | | 85.4 | | | 41.1 | | | 36.4 | | | 6.2 | 21.0 |
| LLaDA | 74.3 | 78.8 | 79.4 | 28.0 | 33.2 | 34.4 | 22.2 | 29.2 | 30.3 | 13.8 | 14.1 | 11.2 |
| LLaDA-MRO-2 | 75.1 | **80.0** | 82.5 | 30.0 | 34.0 | 35.6 | 25.3 | 30.0 | 32.8 | 16.7 | 16.9 | 16.0 |
| LLaDA-MRO-4 | **76.9** | 79.6 | **82.6** | **31.0** | **34.2** | **36.2** | **26.3** | **32.1** | **34.3** | **21.4** | **22.0** | **17.2** |
| LLaDA-s1 | 70.7 | 76.4 | 78.7 | 25.0 | 26.0 | 28.8 | 21.7 | 25.3 | 28.3 | 10.2 | 12.4 | 8.4 |
| LLaDA-s1-MRO-2 | 71.6 | 76.7 | 79.6 | **28.2** | 27.6 | **30.0** | 22.7 | 27.3 | 30.3 | 14.5 | 15.2 | 12.0 |
| LLaDA-s1-MRO-4 | **73.3** | **77.7** | **80.1** | 27.8 | **28.0** | 29.0 | **24.2** | **29.3** | **32.8** | **17.3** | **17.8** | **15.2** |

Table 1: Accuracies (%) on mathematical reasoning, scientific reasoning, and logical reasoning tasks. The best performance in each group is highlighted in **bold**. The suffixes "-2" and "-4" indicate the beam size $k$ used during rejection sampling, set to 2 and 4, respectively. Note that for GSM8K and MATH500, we use 4-shot, while for GPQA, Countdown, and Sudoku, we use 5-shot. † indicates that the results for GSM8K and GPQA are taken from [23].

In the experiments presented in this subsection, for a given sampled sequence, we randomly selected two consecutive denoising steps for optimization. The block length was set to 64 for Countdown and Sudoku. Further training details can be found in Appendix C.

**Results.** The experimental results are listed in Table 1. From the results, we observe that rejection sampling with MRO can effectively enhance the reasoning capabilities of models across various types of reasoning tasks. This further validates that our MRO approach is able to adjust model parameters to better capture token correlations during the decoding process, leading to improved reasoning performance. We also observe that our approach can yield particularly significant improvements in logical reasoning tasks, such as Countdown and Sudoku. Notably, based on the LLaDA model, MRO-4 achieves an improvement of +7.9 points on the Countdown task when the response length is set to 128. This is because these tasks seem to rely more heavily on token correlations, which may explain the more pronounced performance gains. Excitingly, the results show that MRO helps DLMs approach the performance of the strong LLM, Qwen2.5-7B, in mathematical reasoning. For example, on the GSM8K task, our model with MRO achieves an accuracy of 82.6%, which is very close to Qwen2.5-7B's 85.4%, indicating that MRO has the potential to narrow the performance gap in mathematical reasoning.

Additionally, we further compare the performance of MRO-2 and MRO-4. In most test cases, MRO-4 can achieve better performance, indicating that expanding the search space by increasing the beam size allows the model to explore a broader range of denoising sequences. However, this comes with a significant computational cost due to the increased sampling requirements. This motivates future research into methods for reducing the computational burden associated with DLM training while still ensuring the ability to search for more optimal denoising sequences.

## 5.5 Analysis

**Decoding Efficiency Analysis.** We present experimental results comparing decoding efficiency under a test-time scaling setting. Our results demonstrate that MRO is computationally efficient, introducing no noticeable decoding burden while significantly improving generation accuracy. Specifically, we provide a de-

| Method | MATH500 | | GPQA | |
|---|---|---|---|---|
| | Decoding Time | Score | Decoding Time | Score |
| Vanilla (LLaDA) | 0.35h∼0.39h | 33.2 | 0.16h∼0.21h | 29.2 |
| LLaDA-TTS + CBR | 0.73h∼0.81h | 35.2 | 0.27h∼0.34h | 30.6 |
| LLaDA-TTS + MRO | 0.84h∼0.85h | 36.0 | 0.31h∼0.44h | 34.6 |

Table 2: Decoding time and performance comparison on MATH500 and GPQA benchmarks.

tailed comparison of decoding time and performance between vanilla decoding and MRO, as summarized in Table 2. Since test-time scaling inherently involves additional sampling, which introduces time overhead, we further compare against a confidence-based reward test-time scaling baseline (denoted as *LLaDA-TTS + CBR*) that incurs the same sampling cost as MRO but omits reward computation. From the results, we observe that although reward computation introduces a modest

amount of additional latency, the overhead remains within a reasonable range and is well justified by the substantial performance gains it yields. Moreover, compared with the confidence-based reward baseline, we find that after excluding the sampling time, our reward design adds only minimal computational overhead while consistently delivering superior performance.

**Quantitative Analysis of Intra- and Inter-sequence Correlations.** We conduct a quantitative analysis of intra- and inter-sequence correlations to better understand our proposed MRO. Specifically, we randomly sampled 200 examples from the MATH500 and GPQA datasets and computed the corresponding intra- and inter-sequence

| Model | MATH500 | | GPQA | |
|---|---|---|---|---|
| | intra-corr | inter-corr | intra-corr | inter-corr |
| LLaDA | 3.44±0.18 | 1.02±0.14 | 2.76±0.21 | 1.02±0.15 |
| LLaDA-MRO | 3.79±0.16 | 1.58±0.12 | 3.34±0.19 | 1.27±0.13 |

Table 3: Intra- and inter-correlation analysis on MATH500 and GPQA benchmarks.

reward scores across five decoding runs with different random seeds. As shown in Table 3, MRO can consistently enhance both intra- and inter-sequence correlations, indicating that it strengthens token correlations within each denoising step and promotes more coherent transitions across steps.

**Performance with Different Denoising Step Sizes.** With a fixed response length of 512, we further explore the performance of LLaDA and LLaDA-MRO-4 with different denoising step sizes. Specifically, we test denoising step sizes of {64, 128, 256, 512}. The results are summarized in Figure 4. From the results, we observe that our model outperforms the baseline across all denoising step sizes. Notably, we find that MRO helps accelerate the denoising process. For example, LLaDA-MRO-4 achieves performance comparable to the baseline with 256 denoising steps, even when using only 128 steps. We attribute this to the inter-sequence correlation rewards we designed during optimization, which encourage the model to consider the collaboration of different steps and facilitate a faster and more efficient denoising process.

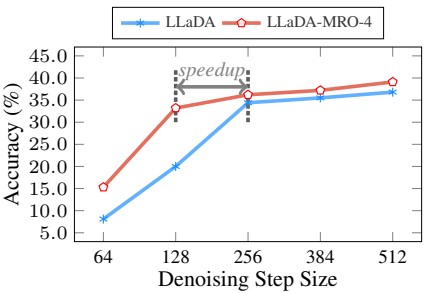

Figure 4: Performance of MRO with different denoising step sizes.

**Performance with Different Temperature Settings.** When optimizing DLMs with our MRO approach, we employ temperature-based sampling to generate diverse responses at each denoising step. Here, we investigate the performance of MRO with different temperature settings. Specifically, we perform test-time scaling on LLaDA and LLaDA-s1 with temperature coefficients of {0, 0.25, 0.5, 0.75, 1.0}. We evaluate the results on the GSM8K task with a length of 256, as shown in Figure 5. From these results, we can observe that a temperature coefficient of 0.25 achieves the best performance. Similar trends can be observed for other reasoning tasks as well. Therefore, in this work, we select a temperature of 0.25 for optimizing the denoising steps.

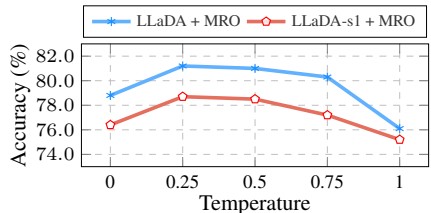

Figure 5: Performance of MRO with different temperature settings.

## 6   Conclusion

In this paper, we introduced the Multi-Reward Optimization (MRO) approach to enhance reasoning in diffusion language models. Specifically, to address the fundamental limitation of token correlation across denoising steps, we designed intra-sequence and inter-sequence correlation rewards, optimized through test-time scaling, rejection sampling, and reinforcement learning. Our experiments show that MRO can significantly improve reasoning, as evidenced by its performance across multiple reasoning benchmarks. Furthermore, we proposed a step-wise group reward optimization approach to tackle reward variance during the optimization process, ensuring efficient optimization. We have demonstrated the effectiveness of this approach through both theoretical analysis and experiments. Our codebase could be found at `https://github.com/wangclnlp/MRO`.

## Acknowledgments

This work was supported in part by the National Science Foundation of China (Nos. 62276056 and U24A20334), the Yunnan Fundamental Research Projects (No.202401BC070021), the Yunnan Science and Technology Major Project (No. 202502AD080014), the Fundamental Research Funds for the Central Universities (Nos. N25BSS054 and N25BSS094), and the Program of Introducing Talents of Discipline to Universities, Plan 111 (No.B16009).

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

# Supplementary Materials for MRO

## A   Limitations and Ethics Statement

### A.1   Limitations

We propose the step-wise group reward optimization (SGRO) approach, which can effectively reduce the reward variance during the reward optimization. However, this introduces new hyperparameters to our MRO, including the size of denoising steps $w$ per group and the size of groups chosen for optimizing DLMs in the rejection sampling and reinforcement learning. The selection of these hyperparameters requires careful consideration. To address this limitation, we conduct comprehensive ablation experiments, as shown in Figure 7. These experimental results provide valuable guidance for selecting the optimal hyperparameters. Additionally, in future work, we will consider designing techniques to determine these hyperparameters automatically.

### A.2   Ethics Statement

This work does not involve any ethical concerns. All data collected for training our DLMs through rejection sampling and reinforcement learning are sourced exclusively from open-source materials. Additionally, this paper may reference certain case study content. However, these references are presented in an elliptical manner, and any potentially harmful content will not be explicitly presented.

## B   Proofs for Theoretical Results

In this section, we provide the proofs for three theoretical results. The first result shows that introducing a potential-based shaping in reward optimization leads to a higher reward variance. The second result demonstrates that using SGRO can mitigate this issue. The third result proves that our token verification reward encourages DLMs to generate responses with stronger internal sequence correlation.

**Property 1.** *Under potential-based reward shaping, the expected reward remains the same, i.e.,* $\mathbb{E}[R(s,a)] = \mathbb{E}[\hat{R}(s,a)]$, *but* $\text{Var}(R(s,a))$ *is higher than* $\text{Var}(\hat{R}(s,a))$.

**Proof:** Let's denote the original reward as $\hat{R}(s_t, a_t)$ and the potential function as $\Phi(s_t)$. The potential-based reward shaping is given by

$$R(s_t, a_t) \quad = \quad \hat{R}(s_t, a_t) + \lambda\Phi(s_{t+1}) - \Phi(s_t) \tag{10}$$

where $\lambda$ is the discount factor. The expected reward under the potential-based shaping is

$$\mathbb{E}[R(s_t, a_t)] \quad = \quad \mathbb{E}[\hat{R}(s_t, a_t) + \lambda\Phi(s_{t+1}) - \Phi(s_t)] \tag{11}$$

$$= \quad \mathbb{E}[\hat{R}(s_t, a_t)] + \lambda\mathbb{E}[\Phi(s_{t+1})] - \mathbb{E}[\Phi(s_t)] \tag{12}$$

Since the expectation of the potential function at the next state $\mathbb{E}[\Phi(s_{t+1})]$ is equal to the expectation of the potential function at the current state $\mathbb{E}[\Phi(s_t)]$ (because the potential function is a function of the state and the state transitions are Markovian) [49], we have

$$\mathbb{E}[R(s_t, a_t)] \quad = \quad \mathbb{E}[\hat{R}(s_t, a_t)] + \lambda\mathbb{E}[\Phi(s_t)] - \mathbb{E}[\Phi(s_t)] \tag{13}$$

$$= \quad \mathbb{E}[\hat{R}(s_t, a_t)] + (\lambda - 1)\mathbb{E}[\Phi(s_t)] \tag{14}$$

For the variance, we have

$$\text{Var}(R(s_t, a_t)) \quad = \quad \text{Var}(\hat{R}(s_t, a_t) + \lambda\Phi(s_{t+1}) - \Phi(s_t)) \tag{15}$$

$$= \quad \text{Var}(\hat{R}(s_t, a_t)) +$$
$$\underbrace{\text{Var}(\lambda\Phi(s_{t+1})) + \text{Var}(-\Phi(s_t)) + 2\text{Cov}(\hat{R}(s_t, a_t), \lambda\Phi(s_{t+1})) +}_{}$$
$$\underbrace{2\text{Cov}(\hat{R}(s_t, a_t), \Phi(s_t)) - 2\text{Cov}(\lambda\Phi(s_{t+1}), \Phi(s_t))}_{\text{additional variance introduced by the potential function terms}} \tag{16}$$

**Algorithm 1** Simplified Grouped Reward Optimization (SGRO)

---

**Input:** Pre-trained DLM $\pi_\theta$; Group Size $w$; Discount $\lambda$; Learning Rate $\eta$; Response Length $L$
**Output:** Fine-tuned DLM $\pi_{\theta^*}$

**for** each prompt $p_0$ in dataset **do**
    **% Roll-out full denoising trajectory**
    $r_T \leftarrow \text{FULLMASK}(L)$
    $states \leftarrow [\,], \ actions \leftarrow [\,]$
    **for** $t \leftarrow T$ **downto** 1 **do**
        sample $r_{t-1} \sim \pi_\theta(\cdot \mid p_0, r_t)$
        $states.append((p_0, r_t, t))$                             $\triangleright s_t$
        $actions.append(r_{t-1})$                               $\triangleright a_t$
    **end for**
    **% SGRO: compute rewards and returns per group**
    $grad \leftarrow 0$
    **for** $g \leftarrow 0$ ; $g < T$ ; $g \leftarrow g + w$ **do**
        $start \leftarrow g, \ end \leftarrow \min(g{+}w, T)$
        $R_{\text{intra}} \leftarrow 0$                                        $\triangleright$ Intra-sequence rewards
        **for** $k \leftarrow start$ **to** $end{-}1$ **do**
            $R_{\text{intra}} \leftarrow R_{\text{intra}} + \text{TOKENVERIFICATIONREWARD}(p_0, actions[k])$
            $R_{\text{intra}} \leftarrow R_{\text{intra}} + \text{PERPLEXITYREWARD}(actions[k])$
        **end for**
        $R_q \leftarrow 0$                                       $\triangleright$ Inter-sequence reward (if final group)
        **if** $end = T$ **then**
            $R_q \leftarrow \text{TASKACCURACYANDFORMATREWARD}(actions[end{-}1])$
        **end if**
        $\phi_{\text{start}} \leftarrow \text{POTENTIALFUNCTION}(states[start])$           $\triangleright$ Potential-based shaping
        $\phi_{\text{end}} \leftarrow \text{POTENTIALFUNCTION}(states[end{-}1])$
        $shaping \leftarrow \lambda \cdot \phi_{\text{end}} - \phi_{\text{start}}$
        $R_{\text{group}} \leftarrow R_{\text{intra}} + R_q + shaping$                     $\triangleright$ Group return
        **for** $k \leftarrow start$ **to** $end{-}1$ **do**
            $\log p \leftarrow \log \pi_\theta(actions[k] \mid states[k])$
            $grad \leftarrow grad + \nabla_\theta(\log p) \cdot R_{\text{group}}$     $\triangleright$ REINFORCE update for steps in this group
        **end for**
    **end for**
    **% Parameter update**
    $\theta \leftarrow \theta + \eta \cdot grad$
**end for**

---

Since $\Phi(s_t)$ and $\Phi(s_{t+1})$ are not independent, the covariance terms are not zero. However, the variance of the potential function terms could add to the variance of the original reward, leading to a higher overall variance. In the general reinforcement learning, [50] has also proven that this result holds when $\Phi(s) > 2\hat{R}(s_t, a_t)$ for all $a$. Furthermore, [54] suggests that such a condition is indeed possible, even when using an optimal baseline reward technique. Therefore, we have

$$\text{Var}(R(s_t, a_t)) > \text{Var}(\hat{R}(s_t, a_t)) \tag{17}$$

This completes the proof of Property 1.

**Property 2.** *Using step-wise group reward optimization can reduce the reward variance, i.e.,* $\text{Var}(R(s, a)) > \text{Var}(R^{(w)}(s, a))$, *where* $R^{(w)}(\cdot)$ *represents the reward after applying step-wise group reward optimization.*

**Proof:** SGRO groups $w$ denoising steps together and provides a reshaped reward for each group. This means that the number of times the potential function is computed and added to the reward is reduced. Let's denote the reward for a group of $w$ steps as

$$R^{(w)}(s_t, a_t) = \sum_{i=0}^{w-1} \hat{R}(s_{t+i}, a_{t+i}) + \lambda\Phi(s_{t+w}) - \Phi(s_t) \tag{18}$$

The variance of this group reward is

$$\text{Var}(R^{(w)}(s_t, a_t)) = \text{Var}\Big( \sum_{i=0}^{w-1} \hat{R}(s_{t+i}, a_{t+i}) + \lambda\Phi(s_{t+w}) - \Phi(s_t) \Big) \tag{19}$$

Following the derivation in Eq. 16, we obtain

$$\begin{aligned}
\text{Var}(R^{(w)}(s_t, a_t)) = {} & \text{Var}(\hat{R}(s_t, a_t)) + \\
& \lambda^2\text{Var}(\Phi(s_{t+w})) + \text{Var}(\Phi(s_t)) + 2\lambda\text{Cov}(\hat{R}(s_t, a_t), \Phi(s_{t+w})) - \\
& 2\text{Cov}(\hat{R}(s_t, a_t), \Phi(s_t)) - 2\lambda\text{Cov}(\Phi(s_{t+w}), \Phi(s_t))
\end{aligned} \tag{20}$$

Compared to Eq. 16, we can observe that due to the larger interval $w$ between $s_{t+w}$ and $s_t$, the correlation between $\Phi(s_{t+w})$ and $\Phi(s_t)$ is typically weaker than the correlation between $\Phi(s_{t+1})$ and $\Phi(s_t)$. Therefore, we have

$$\big|\text{Cov}(\Phi(s_{t+w}), \Phi(s_t))\big| < \big|\text{Cov}(\Phi(s_{t+1}), \Phi(s_t))\big| \tag{21}$$

Here, the negative term $-2\lambda\text{Cov}(\Phi(s_{t+w}), \Phi(s_t))$ in the variance expression of $R^{(w)}(s_t, a_t)$ has a larger absolute value compared to $-2\lambda\text{Cov}(\Phi(s_{t+1}), \Phi(s_t))$ in the variance expression of $R(s_t, a_t)$, as the correlation between $\Phi(s_{t+w})$ and $\Phi(s_t)$ is weaker. Therefore, we can obtain that this results in a smaller overall variance for $R^{(w)}(s_t, a_t)$:

$$\text{Var}(R^{(w)}(s_t, a_t)) < \text{Var}(R(s_t, a_t)) \tag{22}$$

This completes the proof of Property 2. Here, one problem may arise when only a single group provides a potential-based reward, as the reward that could guide the model toward better token correlation may be lost. Therefore, in this approach, we strike a balance between the reward signal and the reward variance for our MRO. The implementation of SGRO is simple, as illustrated in Algorithm A.

**Property 3.** *When a DLM is optimized to achieve higher token verification rewards, it consequently enhances its intra-sequence correlation.*

**Proof:** We provide a theoretical proof for the token verification reward by analyzing it from the perspective of mutual information. Firstly, we recall the definition of the token verification reward at the denoising step $t$:

$$R_t^{\text{tv}} = \frac{1}{N} \sum_{n=1}^{N} \text{Pr}_\theta(r_t^{m_n} \mid p_0, r_{t-1}/r_{t-1}^{m_n}) \tag{23}$$

where $M = \{m_1, \ldots, m_N\}$ is the set of masked token indices, $r_{t-1}^M$ is the set of the predicted tokens at these positions, and $r_{t-1}/r_{t-1}^M$ is the set of unmasked token set. The joint probability over the masked positions can be factorized autoregressively (within the masked set) as:

$$\text{Pr}_\theta(r_{t-1}^M \mid p_0, r_t) = \prod_{n=1}^{N} \text{Pr}_\theta(r_{t-1}^{m_n} \mid p_0, r_{t-1}/r_{t-1}^M, r_{t-1}^{<m_n}) \tag{24}$$

where the rest of the sequence is fixed and only the masked tokens are predicted. The token verification reward approximates this joint modeling by computing leave-one-out log-probabilities:

$$R_t^{\text{tv}} = \frac{1}{N} \sum_{n=1}^{N} \log \text{Pr}_\theta(r_t^{m_n} \mid p_0, r_{t-1}/r_{t-1}^{m_n}) \tag{25}$$

Now define the empirical average pairwise mutual information (PMI) among the masked tokens:

$$\text{PMI}_{\text{avg}} = \frac{2}{N(N-1)} \sum_{1 \leq i < j \leq N} \text{I}(r_{t-1}^{m_i}; r_{t-1}^{m_j} \mid p_0, r_{t-1}/r_{t-1}^M) \tag{26}$$

| Model/Length | GSM8K | | | MATH500 | | | GPQA | | | Countdown | | Sudoku |
|---|---|---|---|---|---|---|---|---|---|---|---|---|
| | 128 | 256 | 512 | 128 | 256 | 512 | 128 | 256 | 512 | 64 | 128 | 64 |
| LLaDA | 74.3 | 78.8 | 79.4 | 28.0 | 33.2 | 34.4 | 22.2 | 29.2 | 30.3 | 13.8 | 14.1 | 11.2 |
| LLaDA-MRO-RS | 76.9 | 79.6 | **82.6** | 31.0 | 34.2 | 36.2 | 26.3 | 32.1 | 34.3 | 21.4 | 22.0 | 17.2 |
| LLaDA-MRO-RL | **77.1** | **80.9** | 81.8 | **33.4** | **35.2** | **37.4** | **28.8** | **33.8** | 33.8 | **24.6** | **27.2** | **20.2** |
| LLaDA-s1 | 70.7 | 76.4 | 78.7 | 25.0 | 26.0 | 28.8 | 21.7 | 25.3 | 28.3 | 10.2 | 12.4 | 8.4 |
| LLaDA-s1-MRO-RS | **73.3** | **77.7** | **80.1** | **27.8** | **28.0** | 29.0 | **24.2** | **29.3** | **32.8** | **17.3** | **17.8** | **15.2** |
| LLaDA-s1-MRO-RL | 71.8 | 75.9 | 78.0 | 26.6 | 27.2 | **29.4** | 23.7 | 28.3 | 32.3 | 17.1 | 16.2 | 13.8 |

Table 4: Results of the MRO in reinforcement learning. The suffixes "-RS" and "-RL" denote the results obtained using rejection sampling with $k = 4$ and reinforcement learning, respectively.

Using a standard second-order Taylor expansion around the independence assumption (as in energy-based models), this quantity can be approximated as:

$$\text{PMI}_{\text{avg}} \approx \frac{2}{N(N-1)} \sum_{i<j} \log \frac{\text{Pr}_\theta(r_{t-1}^{m_i}, r_{t-1}^{m_j} \mid \cdot)}{\text{Pr}_\theta(r_{t-1}^{m_i} \mid \cdot)\text{Pr}_\theta(r_{t-1}^{m_j} \mid \cdot)} \tag{27}$$

Importantly, the leave-one-out log-probabilities computed in TVR serve as sufficient statistics for estimating these pairwise interactions. Therefore, maximizing the average leave-one-out log-probability is first-order equivalent to maximizing $\text{PMI}_{\text{avg}}$, thereby promoting stronger intra-sequence correlation. This completes the proof of Property 3.

## C   Experiments

In this section, we provide additional experimental details and present the experimental results of MRO with reinforcement learning.

### C.1   Experimental Details

**Training Setups.**   For training LLaDA-s1, we used a pre-trained version of LLaDA. The learning rate was set to 2e-5. We trained this model on the s1 dataset for 3 epochs. In contrast to [26], we found that training for more epochs on the s1 dataset did not result in further performance improvements. For rejection sampling and reinforcement learning, we set the learning rate to 2e-6. During training, we performed model validation every 50 steps and selected the best model based on performance on the validation set as our final model. For computing $R_t^{\text{tv}}$, we sampled one token from the predicted masked tokens at each denoising step. For $R_t^{\text{ppl}}$, we set $C_{\text{ppl}}$ and $F_{\text{ppl}}$ to 100 and 100, respectively. In Figure 1, the x-axis corresponds to the intra- and inter-sequence correlation rewards defined in Section 4.2: the left part presents the cumulative intra-sequence rewards $R_t^{\text{tv}} + R_t^{\text{ppl}}$, while the right part shows the inter-sequence reward $R_0^q$. As the reward scales differ, we applied standardization for visualization.

**Training Datasets.**   For both rejection sampling and reinforcement learning, we utilized DeepScaleR [55] in conjunction with the 10k Countdown[5] and Sudoku[6] datasets. These datasets were randomly shuffled to ensure a well-balanced data distribution.

**Evaluation.**   For the evaluation, we focus on five reasoning tasks: GSM8K [56], MATH500 [57], GPQA [58], Countdown[7], and Sudoku[8]. During testing, we set the sampling temperature to 0.25.

### C.2   Reinforcement Learning

We explore the use of reinforcement learning to implement our MRO. The experimental setup and results are presented below.

---

[5] https://huggingface.co/datasets/Jiayi-Pan/Countdown-Tasks-3to4

[6] https://huggingface.co/datasets/Ritvik19/Sudoku-Dataset

[7] https://github.com/HKUNLP/diffusion-vs-ar

[8] https://github.com/dllm-reasoning/d1/blob/main/dataset/4x4_test_sudoku.csv

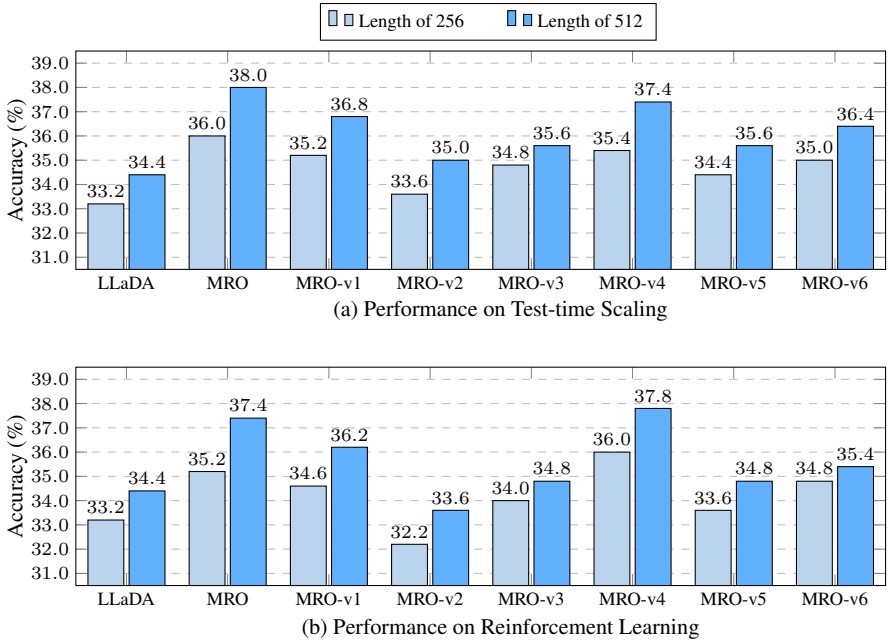

Figure 6: Performance comparison of different MRO variants on the MATH500 benchmark for both test-time scaling and reinforcement learning.

**Task Setup.** We employed the REINFORCE [59] to perform this optimization. Specifically, during the optimization process, we used a temperature-based sampling to obtain a denoising sequence $\{\hat{r}_T, \hat{r}_{T-1}, \cdots, \hat{r}_0\}$. Subsequently, we use a REINFORCE to train our DLMs through a cumulative reward. The loss function can be given by

$$\mathcal{L}_{\text{rl}}(\theta) = -\mathbb{E}_{o_t, p_0, \{\hat{r}_T, \hat{r}_{T-1}, \cdots, \hat{r}_0\}} \left[ \frac{1}{o_t} \sum_{i=1}^{L_{\hat{r}}} \mathbf{1}[\hat{r}_t^i = M] \log \Pr_\theta(\hat{r}_{t-1}^i | p_0, \hat{r}_t) \right] R_{\text{acc}} \quad (28)$$

where $R_{\text{acc}}$ is the cumulative reward, computed as: $R_{\text{acc}} = \sum_{t=T}^{0} R(s_t, a_t)$. Similarly, we apply the SGRO in reinforcement learning, where we group the sampled sequence, with each group containing $w$ steps. For each group, we provide a reshaped reward. Additionally, during the optimization, similar to rejection sampling, instead of using all groups, we sample only a subset of the groups. Building upon this, we integrated quality evaluation scores into the reinforcement learning training process through a shaping mechanism. Specifically, we selected a group $\hat{r}_{i:i+w}$. Then, we computed the quality rewards $R_i^q$ and $R_{i+w}^q$ for the first and last steps of the group, respectively. The final quality reward for the group was determined by the difference between these two quality rewards. Although we implemented MRO in the DLM using REINFORCE, any reinforcement learning algorithm (such as PPO [39] or GRPO [60]) could be used to achieve MRO, as outlined in the modeling framework presented in Section 4.2.3. The primary focus of this work is to demonstrate the effectiveness of our MRO in enhancing token correlation, rather than to explore the performance of different reinforcement learning algorithms in the DLM training process. Therefore, we did not conduct tests of these algorithms one by one.

**Results.** We compare the performance of our DLMs trained via reinforcement learning to those trained with rejection sampling and the LLaDA instruction model. The results are listed in Table 4. From the results, we observe that with reinforcement learning, our MRO still achieves significant improvements across various benchmarks. However, when compared to rejection sampling, our reinforcement learning approach does not show a substantial advantage. We identify two main reasons for this observation. First, our rejection sampling approach has been enhanced compared to the original; it combines offline data construction with online sampling techniques. Second, our reinforcement learning approach is relatively basic and does not incorporate advanced modifications or improvements.

## C.3 Ablation Study

In this subsection, we design several MRO variants to further describe the functionality of intra-sequence and inter-sequence rewards, as well as SGRO.

**Reward Design.** As shown in Table 5, we design six MRO variants to investigate the impacts of intra-sequence and inter-sequence rewards.

| Reward\Variant | MRO-v1 | MRO-v2 | MRO-v3 | MRO-v4 | MRO-v5 | MRO-v6 |
|---|---|---|---|---|---|---|
| $R_t^{\text{tv}}$ | ✓ | | | ✓ | | ✓ |
| $R_t^{\text{ppl}}$ | | ✓ | | | ✓ | ✓ |
| $R_0^{\text{q}}$ | | | ✓ | ✓ | ✓ | |

Table 5: Description of MRO variants.

We conduct experiments on test-time scaling and reinforcement learning to evaluate the performance of these variants. As illustrated in Figure 5, the results reveal that achieving superior performance with a single reward combination is challenging. For example, MRO-v2 and MRO-v3 demonstrate relatively poor performance compared to other variants in both test-time scaling and reinforcement learning. Furthermore, the token verification reward proves to be highly effective. We can see that MRO-v2, MRO-v3, and MRO-v5 perform worse in comparison. However, when compared to LLaDA, it is clear that all of our reward designs are effective, except for MRO-v2 in reinforcement learning, which exhibits some performance degradation. Other variants, to varying degrees, lead to performance improvements. This further supports the correctness of our design approach, centered around enhancing token correlation through tailored reward strategies. Interestingly, results in reinforcement learning show that MRO-v4 outperforms MRO itself. This improvement could be attributed to sampling and the potential reward variance caused by the perplexity reward. Nevertheless, since other experiments have confirmed the usefulness of this reward, we chose not to discard it.

**Step-wise Group Reward Optimization.** We also conduct an ablation study on our SGRO. Specifically, we test the case where SGRO is not applied, meaning that each denoising step receives a shaping reward in rejection sampling. As shown in Table 6, we find that SGRO is effective and helps the MRO achieve better performance.

# D Analysis

## D.1 Comparison of MRO with Other Reasoning-Enhanced DLM Approaches

We compare MRO with other existing reasoning-enhanced models and approaches. These include DiffuLLaMA [34], which adapts the LLaMA model for DLMs; EDLM [35], which introduces an energy function to enhance sequence-level correlation; Dream [24], which utilizes the Qwen-2.5-7B model for initialization; and d1-LLaDA [26], which trains the LLaDA-8B-Instruct model using GRPO. The results are presented in Table 7. Note that, except for EDLM, the results for the other baselines are taken directly from the original papers. For EDLM, we replicate its autoregressive energy function version in our codebase. First, compared to DiffuLLaMA and Dream, our MRO achieves competitive results. Moreover, we observe that although EDLM incorporates sequence-level correlation (i.e., inter-sequence correlation

| Model/Length | GSM8K | | MATH500 | |
|---|---|---|---|---|
| | 256 | 512 | 256 | 512 |
| DiffuLLaMA | 63.1 | | - | - |
| Dream-7B-Instruct | 77.2 | | - | - |
| EDLM | 78.1 | 80.0 | 34.4 | 35.4 |
| d1-LLaDA | 81.1 | 82.1 | 38.6 | 40.2 |
| LLaDA-MRO-RS | 79.6 | 82.6 | 34.2 | 36.2 |
| LLaDA-MRO-RL | 80.9 | 81.8 | 35.2 | 37.4 |
| LLaDA-MRO-TS | 82.5 | 82.9 | 39.4 | 42.6 |

Table 7: Performance comparison of MRO with other reasoning-enhanced models and approaches. "-TS" indicates that we use task-specific training data to optimize the DLM in reinforcement learning.

described in this paper), our MRO still outperforms it. We attribute this to the lack of consideration for intra-sequence correlation in EDLM. However, we observe that our model performs slightly worse than the d1-LLaDA model. We argue that this comparison is not entirely fair, as d1-LLaDA is trained using a task-specific training set for GRPO. This potentially gives it an advantage by benefiting from

| Model/Length | MATH500 | | GPQA | |
|---|---|---|---|---|
| | 256 | 512 | 256 | 512 |
| LLaDA-MRO | **34.2** | **36.2** | **32.1** | **34.3** |
| LLaDA-MRO w/o SGRO | 32.8 | 35.4 | 31.3 | 33.3 |

Table 6: Ablation study of SGRO.

task-specific training. To validate this, we also conduct reinforcement learning using a task-specific training set. More specifically, during the reinforcement learning training, we use the GSM8K[9] and MATH500[10] training sets to perform the MRO, respectively. We find that the results from this training approach surpass the performance of d1-LLaDA.

## D.2 Scaling Training with Different Group Sizes

We scale training in MRO by using different group sizes. Specifically, we keep the total number of training samples constant while testing various group sizes {1, 2, 4, 8, 16, 32, 64} during both rejection sampling and reinforcement learning. The results are shown in Figure 7. From the results, we can find that increasing the group size provides some benefits, but these benefits diminish after a group size of 2, with minimal improvements beyond that. Notably, performance becomes

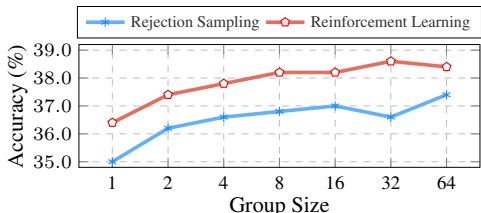

Figure 7: Performance of MRO with different temperature settings, evaluated on the MATH500 benchmark.

unstable after a group size of 16. Considering both performance and training costs, we choose a group size of 2 for our experiments.

## D.3 Performance with Different Perplexity Reward Upper Bounds

We conduct an ablation study by varying the upper bound of the perplexity reward over the range {50, 80, 100, 130, 150}, implemented on the LLaDA-8B-Instruct model. We use the LLaDA-8B-Instruct model as the baseline. The results show that model performance remains relatively stable when the bound lies between 80 and 130, demonstrating the robustness of our method as well as the insensitivity of this hyperparameter. In contrast, a smaller bound tends to compress reward differences and weaken the learning

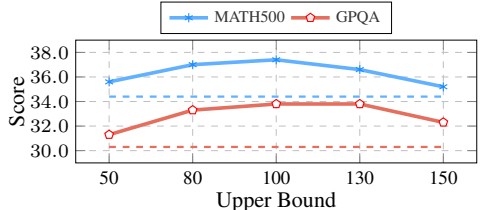

Figure 8: Performance with different perplexity reward upper bounds. The solid line denotes the performance of MRO, while the dashed line represents the baseline.

signal, whereas a larger bound introduces excessive reward variance, both of which can negatively affect training stability.

---

[9] https://huggingface.co/datasets/openai/gsm8k
[10] https://github.com/openai/prm800k/tree/main/prm800k/math_splits

| | |
|---|---|
| **Prompt** | Kylar went to the store to buy glasses for his new apartment. One glass costs $5, but every second glass costs only 60% of the price. Kylar wants to buy 16 glasses. How much does he need to pay for them? |
| **Answer** | 64 |
| **LLaDA** | To determine how much Kylar needs to pay for 16 glasses, we need to consider the cost of the first glass and the discounted of every second glass.

1. The first glass costs $5.
2. Every second glass costs 60% of the price price of the first glass, which is $16 \times 0.60 = 9.60$ dollars.

Let's calculate the total cost:

- The first glass costs $5.
- The second glass costs $9.60.
- The third glass costs $5.
- The fourth glass costs $9.60.
- $\cdots$
- The sixteenth glass costs $5.

So, the total cost is:
5 + 9.60 + 5 + 9.60 + 5 + 9.60 + 5 + 9.60 + 5 + 9.60 + 5 + 9.60 + 5 + 9.60 + 5 + 9.60 + 5 = 120

The answer is $120. |
| **LLaDA + MRO** | To determine how much Kylar needs to pay for 16 glasses, we need to consider the cost of the first glass and the discounted of the second glass.

1. The cost of the first glass is $5.
2. The cost of the second glass is 60% of the first glass's price, which is $0.6 \times 5 = 3$ dollars.

Since Kylar wants to buy 16 glasses, he can pair them up follows:
- First glass: $5
- Second glass: $3
- Third glass: $5
- Fourth glass: $3
- $\cdots$
- Sixteenth glass: $5

Each pair of glasses consists of $5 + $3 = $8.

There are 8 pairs of glasses (since 64 divided by 2 is 8).

So, the total cost is:
8 pairs $\times$ 8 per pair = $64

Therefore, Kylar needs to pay $64 for 16 glasses.

The answer is $64. |

Table 8: A case from the test-time scaling experiments. MRO provides accurate guidance for token correlation and helps identify a more precise CoT-style reasoning path. Note that we remove the "<think></think>" and "<answer></answer>" format tags to improve the readability of this case.

| | |
|---|---|
| **Prompt** | Two sides of a triangle are each 8 units long. If the third side has a whole number length, what is the greatest possible perimeter, in units, for the triangle? |
| **Answer** | 31 |
| **LLaDA** | $x$ is a positive number, this condition is always satisfied

From conditions 1 and 2, we get:

$$16 > x$$
$$x < 0$$

Since $x$ must be a whole number, the largest possible value for $x$ is satisfies both conditions is $x = 7$.

Now, we calculate the perimeter:
$$\text{Perimeter} = 8 + 8 + 7 = 23$$

Thus, the greatest possible perimeter of the triangle is 23. |
| **LLaDA-MRO-RS** | To solve:

1. We know to form a triangle, the sum of the lengths of any two sides must be greater than the length of the third side.
2. Let the denote the length of the third side as $x$.
3. We to satisfy the triangle inequality theorem:
- $8 + 8 > x$
- $8 + x > 8$
- $x + 8 > 8$

4. Simplifying these inequalities:
- $16 > x$
- $x > 0$
- $x > 0$

5. Combining these inequalities, we get:
- $0 < x < 16$

6. Since $x$ must be a whole number, the possible values for $x$ are 1, 2, 3, 4, 5, 6, 7, 8, 9, 10, 11, 12, 13, 14, and 15.

7. The find the greatest possible perimeter, we choose the largest possible value for $x$:
- $x = 15$

8. The perimeter of the triangle is the sum of its sides:
- Perimeter $= 8 + 8 + 15 = 31$

Therefore, the greatest possible perimeter for the triangle is 31. |

Table 9: A case from the rejection sampling experiments. We find that using MRO to search for better and more accurate reasoning paths, and then fine-tuning the DLM with these paths, helps the model tend toward better reasoning paths during the decoding process.

| | |
|---|---|
| **Prompt** | Artie has a flower stand at the Farmers Market. He sells three kinds of flowers: marigolds, petunias and begonias. He usually sells marigolds for $2.74 per pot, petunias for $1.87 per pot and begonias for $2.12 per pot. Artie has no change today, so he has decided to round all his prices to the nearest dollar. If Artie sells 12 pots of marigolds, 9 pots of petunias and 17 pots of begonias, how much will he make? |
| **Answer** | 88 |
| **LLaDA** | To determine how much Artie will make, we need to calculate the total revenue from each type of flower and then sum these amounts.

1. **Marigolds:**
- Price per pot: $2.74
- Number of pots sold: 12
- Revenue from marigolds: $2.74 \times 12 = 32.88$

2. **Petunias:**
- Price per pot: $1.87
- Number of pots sold: 9
- Revenue from petunias: $1.87 \times 9 = 16.83$

3. **Begonias:**
- Price per pot: $2.12
- Number of pots sold: 17
- Revenue from begonias: $2.12 \times 17 = 36.04$

Now, we sum the revenues:

$$32.88 + 16.83 + 36.04 = 85.75$$

The answer is $86. |
| **LLaDA-MRO-RL** | First, we's round the prices to the nearest dollar:
- Marigolds: $2.74 rounds to $3
- Petunias: $1.87 rounds to $2
- Begonias: $2.12 rounds to $2

Next, we calculate the total revenue for each type of flower:
- Marigolds: 12 pots * $3/pot = $36
- Petunias: 9 pots * $2/pot = $18
- Begonias: 17 pots * $2/pot = $34

Finally, we add up the revenues together:
$36 + $18 + $34 = $88

The answer is $88. |

Table 10: A case from the reinforcement learning experiments. Interestingly, after applying reinforcement learning for MRO optimization, we find that the DLM generates shorter and more accurate CoT-style reasoning paths. We conjecture that this improvement is enabled by the optimization against the inter-sequence correlation reward, which encourages the model to achieve better format and accuracy rewards with fewer denoising steps.

