# OpenReview forum: "MRO: Enhancing Reasoning in Diffusion Language Models via Multi-Reward Optimization"
_NeurIPS.cc/2025/Conference — NeurIPS 2025 poster_

### Official Review · Reviewer_Sm7r · 2025-06-28

**Clarity:** 2
**Significance:** 2
**Originality:** 3
**Rating:** 3
**Confidence:** 4

**Summary:**

This paper introduces MRO, a novel framework to improve reasoning capabilities in DLMs by directly enhancing token correlation during the denoising process. Empirical results on multiple reasoning benchmarks (e.g., GSM8K, MATH500, GPQA, Countdown, Sudoku) show that MRO significantly improves the reasoning performance of DLMs and even narrows the gap with strong AR LLMs.

**Questions:**

see weakness

**Ethical Concerns:**

["NO or VERY MINOR ethics concerns only"]

**Final Justification:**

Overall, my main concern remains—the theoretical analysis is not sufficient to demonstrate why the approach works or why this design is an effective choice. The authors acknowledge that the reward design is not optimal and defer improvements to future work, which undermines the claimed contribution. After reading the discussion between the authors and other reviewers, I also agree that the structure of the paper is unclear and needs further refinement. Taking everything into consideration, I have decided to keep my score.

**Limitations:**

yes

**Quality:**

3

**Strengths And Weaknesses:**

**Strength:**

The method is tested across a diverse set of reasoning benchmarks and consistently demonstrates improvements over strong DLM baselines. The performance gains are especially pronounced in domains such as logical reasoning (e.g., Sudoku, Countdown), where token correlation is critical.

**Weakness:**

- While the paper introduces a multi-reward framework to enhance token correlation, it lacks a rigorous theoretical foundation linking these rewards directly to improved reasoning performance. The definition of intra- and inter-sequence correlation rewards is intuitive, but the paper does not sufficiently justify why these particular reward signals are optimal.
- The comparisons between DLMs with MRO and autoregressive LLMs like Qwen2.5 and LLaMA-3-8B appear to be superficially favorable but are not fair. What is the setup of AR models? Are they evaluated with the same prompt formats, decoding strategies, or number of shots? Without full control of experimental settings, the claim that MRO narrows the gap with AR models is not strongly substantiated
- MRO introduces considerable computational complexity due to repeated model evaluations, especially for the token verification reward that requires re-inference of multiple masked variants. This paper does not quantify the actual computational overhead compared to baseline models. The reliance on test-time scaling and beam search with reward evaluation also raises concerns about scalability to real-world applications.
- The paper only compare the performance with AR models. They didn't compare MRO with other RL methods.
- The study only looks at STEM tasks; it doesn't assess how well the suggested MRO apply to tasks involving common sense, following instructions, and so on.
- The experiments are limited to models up to 8B parameters, leaving the effectiveness of MRO on larger models (e.g., 13B, 30B, or more) untested.
- The paper only consider the experiments with few-shots. It is unclear about the performance of zero-shot.

---

> ### Author Rebuttal · Authors · 2025-07-31
>
> Dear Reviewer sm7r,
>
> We appreciate the reviewer’s constructive and thoughtful feedback.
>
> We would like to thank the reviewer for the positive feedback regarding "the consistent performance improvements over strong DLM baselines across diverse reasoning benchmarks". We especially appreciate the recognition of "our method’s effectiveness in logic-intensive tasks such as Sudoku and Countdown, where token correlation plays a crucial role".
>
> Below, we explain the main points you are concerned about.
>
> ---
>
> >*W1: The definition of intra- and inter-sequence correlation rewards is intuitive, but the paper does not sufficiently justify why these particular reward signals are optimal.*
>
> Thanks for your insightful feedback! We would like to clarify that the primary goal of our work is to improve DLM reasoning from the perspective of token correlation, rather than to design an optimal reward function. In this work, we formally define the token correlation problem and introduce intuitive formulations of intra-/inter-sequence correlation rewards as a first step toward addressing it. By optimizing these reward signals, we demonstrate that token correlation plays a crucial role in enhancing the reasoning capabilities of DLMs. As also mentioned by Reviewer3ZHw, we believe our contribution serves as a direction-setting effort, and the design of more optimal reward functions remains an open direction for future research.
>
> >*W2: What is the setup of AR models? Are they evaluated with the same prompt formats, decoding strategies, or number of shots?*
>
> Our evaluation setup for AR models follows the configuration reported in the LLaDA paper. We ensure the use of identical prompt formats and number of shots to maintain fair comparisons. The evaluation is conducted using the *lm-evaluation-harness* framework, which provides standard decoding strategies and shot settings for the benchmarks we use. Additionally, the resulting scores are consistent with those reported in prior literature.
>
> >*W3: MRO introduces considerable computational complexity due to repeated model evaluations, especially for the token verification reward that requires re-inference of multiple masked variants.*
>
> Thanks for your insightful comment! Below, we clarify the issue related to inference/training time and present empirical results to support our claims. **Taken together, we demonstrate that our MRO is highly practical and well-suited for real-world applications**.
>
> - Regarding your concern about inference-time overhead, we first would like to clarify that in both the rejection sampling and RL fine-tuning settings, **multi-reward computation and policy gradient updates are performed only during training**. Once the model parameters are updated, decoding proceeds identically to the standard LLaDA baseline \[22], with **no additional reward computation or repeated forward passes involved during inference**.
>
> - Additionally, to further address your concern about the overhead introduced by our reward computation, we provide a detailed breakdown of the associated computational cost in the table below, where $\Theta$ denotes the model size and $L$ denotes the sequence length. To reduce the overall burden, we adopt several efficiency-oriented strategies. Specifically, **we leverage SGRO to sparsely compute rewards across decoding steps, significantly limiting the number of evaluations needed. For the Perplexity Reward, we employ a lightweight model (GPT-2-small) to minimize FLOPs. We also employ GPU-parallelized computation to process multiple reward evaluations concurrently during training, substantially reducing wall-clock latency**.
>
>     |Reward Type| Model Used | Formula (Per Occurrence) | Occurrence Count (w/ SGRO) |Total FLOPs (Estimated)|
>     | - | :-: | :-: | :-: | :-: |
>     |Token Verification Reward| LLaDA-8B  | $2 \cdot \Theta_{\text{LLaDA}} \cdot L \cdot (N + 1)$ | $\frac{T}{w} = \frac{128}{w}$    | $\frac{128}{w} \cdot 1.23 \times 10^{13}$ |
>     |Perplexity Reward| GPT-2-Small  | $2 \cdot \Theta_{\text{GPT-2}} \cdot L$ | $\frac{T}{w} = \frac{128}{w}$    | $\frac{128}{w} \cdot 6.0 \times 10^{10}$  |
>
>
> - We have also conducted a comparison of both inference and training performance with and without our reward design, as shown in the tables below. The runtime was measured on eight A800 GPUs under data parallelism, with each configuration run five times to report a range of values. The results demonstrate that **our reward introduces only a modest computational overhead while delivering substantial performance improvements**, confirming its practicality and effectiveness.
>
>     |Method|Inference Time (MATH500)|Training Time|Score (MATH500)|
>     |-|:-:|:-:|:-:|
>     |LLaDA|0.35h~0.39h|-|33.2|
>     ||
>     |LLaDA-RS w/o MRO|0.35h~0.38h|10.3h~11.0h|33.0|
>     |LLaDA-RS w/ MRO|0.34h~0.37h|12.7h~13.1h|34.2|
>     ||
>     |LLaDA-RL w/o MRO|0.33h~0.37h|8.2h~8.4h|33.4|
>     |LLaDA-RL w/ MRO|0.36h~0.39h|9.6h~10.1h|35.2|
>     ||
>     |LLaDA-TTS w/o MRO|0.73h~0.81h|-|35.2|
>     |LLaDA-TTS w/ MRO|0.84h~0.85h|-|36.0|
>
> >*W4: The paper only compares the performance with AR models. They didn't compare MRO with other RL methods.*
>
> To the best of our knowledge, our work is among the earliest to explore RL for DLMs. In the paper, we have made direct comparisons between MRO and other RL methods in the context of DLMs.
>
> - As of the NeurIPS2025 submission deadline (May 15), the only concurrent work that conducted RL training on DLMs is *d1-LLaDA* (released on arXiv on April 16). In Table 5 (Appendix), we provide a detailed comparison with d1-LLaDA *under the same training settings*. As shown in the results (copied below), our proposed MRO achieves consistently better performance than d1-LLaDA.
>     |Model/Length|GSM8K||MATH500||
>     |-|:-|:-:|:-|:-:|
>     ||256|512|256|512|
>     |d1-LLaDA|81.1|82.1|38.6|40.2|
>     |LLaDA-MRO | **82.5** | **82.9** | **39.4** | **42.6** |
>
> - We also conducted ablation studies using various reward designs. Specifically, we evaluate LLaDA with individual reward signals—namely, LLaDA-$R^{tv}\_{t}$, LLaDA-$R^{ppl}\_{t}$, and LLaDA-$R^{q}\_{0}$, under RL training. These variants can serve as RL baselines to assess the contribution of each reward component. The results are summarized in the table below and further illustrated in Figure 6 (Appendix), with the partial scores copied here for reference (_as indicated in parentheses in the table_). From the results, we observe that our multi-reward design consistently outperforms the single-reward baselines, highlighting the effectiveness of combining multiple reward signals.
>     |Model||Reinforcement Learning||
>     |-|:-:|:-:|:-:|
>     ||MATH500|GPQA|Countdown|
>     |LLaDA-$R^{tv}_{t}$|36.2(+1.8)|32.7(+2.4)|25.3(+11.2)|
>     |LLaDA-$R^{ppl}_{t}$|33.6(-0.8)|30.8(+0.5)|18.9(+4.8)|
>     |LLaDA-$R^{q}_{0}$|34.8(+0.4)|31.2(+0.9)|23.5(+9.4)|
>     |LLaDA + MRO|**37.4(+3.0)**|**33.8(+3.5)**|**27.2(+13.1)**|
>
> >*W5: The study only looks at STEM tasks; it doesn't assess how well the suggested MRO applies to tasks involving common sense, instruction following, and so on.*
>
> Thanks for the thoughtful suggestion! We have extended our evaluation to a broader range of tasks, including general knowledge question answering (MMLU), code generation (HumanEval), and instruction following (AlpacaEval2 and Arena-Hard), to verify the effectiveness of our proposed MRO further. The results demonstrate that **our MRO achieves strong robustness and can deliver consistent improvements (albeit to varying degrees) across these diverse tasks**. The results and details are as follows.
>
> |Model|MMLU|HumanEval|AlpacaEval2|Arena-Hard|
> |-|:-:|:-:|:-:|:-:|
> |LLaMA-3-8B-Instruction|68.4|59.8|25.3|22.3
> |Mistral-7B-Instruct|60.1|30.5|14.7|12.6
> |Deepseek-LLM-7b-Chat|48.2|26.2|11.2|10.3
> |Qwen2.5-7B-Instruct|71.9|56.7|27.8|25.2
> |||
> |LLaDA|65.5 |47.6 |16.3 |10.0 |
> |LLaDA-RS + MRO|67.5 | 48.1 | **20.2** | 12.3 |
> |LLaDA-RL + MRO|**68.2** | **50.0** | 19.4 | **15.7**|
>
> >*W6: The experiments are limited to models up to 8B parameters, leaving the effectiveness of MRO on larger models (e.g., 13B, 30B, or more) untested.*
>
> As you said, "the experiments are limited to models up to 8B parameters", **we would like to clarify that current open-source diffusion language models are only available up to this scale**. Nevertheless, to further demonstrate the robustness of our method, we additionally develop LLaDA-s1, a reasoning-specialized diffusion language model. As shown in Table 1, experimental results show that MRO consistently improves performance even on this customized model.
>
> >*W7: The paper only considers the experiments with a few shots. It is unclear about the performance of zero-shot.*
>
> Thanks for the helpful feedback! In our main experiments, we adopted the same shot setting as our baseline methods to ensure a fair comparison. To address your concern, we have additionally conducted experiments under the zero-shot setting, and the results are provided below. **As shown, our method continues to deliver effective improvements even without in-context demonstrations (i.e., zero-shot), highlighting its robustness across different evaluation regimes**.
>
> |Model/Length|GSM8k (0-shot)||MATH500 (0-shot)||
> |-|:-|:-:|:-|:-:|
> ||256|512|256|512|
> |LLaDA|75.7|78.4|28.2|30.0|
> |LLaDA-MRO-RS|77.5|79.6|29.8|32.4|
> |LLaDA-MRO-RL|**79.2**|**81.0**|**30.2**|**33.2**|
>
>
> ---
> We sincerely thank you for your positive feedback on our paper!
>
> Best,
>
> Authors

---

> ### Author Response · Authors · 2025-08-06
> **Reponse to Reviewer Sm7r (Part 1)**
>
> Dear Reviewer sm7r,
>
> Thank you again for your thoughtful feedback and for continuing to engage with our paper. Below, we address your remaining concerns regarding reward design, RL baseline, and computational overhead.
>
> ---
>
> >*The paper frames intra- and inter-sequence correlation rewards fail to provide any theoretical analysis or empirical ablation to justify why these are the right or even effective choices. ... The work thus lacks both theoretical depth and genuine novelty in reward design.*
>
> Thank you for your feedback. We would like to clarify our response to the concern that *"the intra- and inter-sequence correlation rewards lack theoretical analysis or empirical ablation to justify their effectiveness*".
>
> - *Empirical Ablation*. As shown in Figure 6 of the appendix, we have conducted extensive ablation studies to evaluate the effectiveness of our reward design. Specifically, we tested all combinations of the proposed reward components under both TTS and RL settings. These experiments consistently demonstrate that our **full multi-reward design outperforms all single-reward baselines**, confirming its empirical effectiveness. A subset of these results was also included in the rebuttal for reference.
>
>
> - *Theoretical Analysis*. In addition to the empirical results, we also provide a theoretical justification for one of the core components, the Token Verification Reward, based on its relationship to pairwise mutual information, showing that it approximates intra-sequence correlation.
>
>     Recall the definition of the TVR at the denoising step $t$:
>
>     $
>     R^{\text{tv}}\_{t}= \frac{1}{N}\sum\_{n=1}^{N}\Pr\_{\theta}(r\_{t}^{m\_{n}}\mid p\_{0},r\_{t-1}/\{r\_{t-1}^{m\_{n}}\}),
>     $
>
>     where $M = \{m_1, \dots, m_N\}$ is the set of masked token indices, $r_{t-1}^{M}$ is the set of the predicted tokens at these positions, and $r_{t-1} / r_{t-1}^{M}$ is the set of unmasked token set. The joint probability over the masked positions can be factorized autoregressively (within the masked set) as:
>
>     $
>     \Pr_{\theta}(r_{t-1}^{M} \mid p_0, r_t)
>     = \prod_{n=1}^{N} \Pr_{\theta}(r_{t-1}^{m_n} \mid p_0, r_{t-1} / r_{t-1}^{M}, r_{t-1}^{<m_n}),
>     $
>
>     where the rest of the sequence is fixed and only the masked tokens are predicted.
>
>     TVR approximates this joint modeling by computing **leave-one-out log-probabilities**:
>
>     $
>     R^{\text{tv}}\_{t}= \frac{1}{N} \sum_{n=1}^{N} \log \Pr_{\theta}(r_{t}^{m_n} \mid p_0, r_{t-1} / r_{t-1}^{m_n})
>     $
>
>     Now define the empirical average pairwise mutual information (PMI) among the masked tokens:
>
>     $
>     \mathrm{PMI}\_{\text{avg}} = \frac{2}{N(N-1)} \sum_{1 \le i < j \le N} \mathrm{I}(r_{t-1}^{m_i}; r_{t-1}^{m_j} \mid p_0, r_{t-1} / r_{t-1}^{M})
>     $
>
>     Using a standard second-order Taylor expansion around the independence assumption (as in energy-based models), this quantity can be approximated as:
>
>     $
>     \mathrm{PMI}\_{\text{avg}} \approx \frac{2}{N(N-1)} \sum_{i < j} \log \frac{\Pr_{\theta}(r_{t-1}^{m_i}, r_{t-1}^{m_j} \mid \cdot)}{\Pr_{\theta}(r_{t-1}^{m_i} \mid \cdot) \Pr_{\theta}(r_{t-1}^{m_j} \mid \cdot)}
>     $
>
>     Importantly, the leave-one-out log-probabilities computed in TVR serve as sufficient statistics for estimating these pairwise interactions. Therefore, **maximizing the average leave-one-out log-probability is first-order equivalent to maximizing $\mathrm{PMI}_{\text{avg}}$, thereby promoting stronger intra-sequence correlation**. We promise to add the theoretical grounding in the revision.

---

> ### Author Response · Authors · 2025-08-06
> **Reponse to Reviewer Sm7r (Part 2)**
>
> >*There is no comparison with a broader set of RL methods or alternative reward functions.*
>
> Thank you for the valuable comment. In addition to the results presented in our initial rebuttal (including baselines using individual reward components such as $R^{\text{tv}}_t$ and $R^{\text{ppl}}_t$), we have further incorporated a broader set of RL methods and reward functions to better evaluate the generality and effectiveness of our proposed approach. These new baselines include:
>
> * **RM-Baseline-1**: A REINFORCE-based method using rule-based rewards derived from format correctness and answer accuracy, following prior setups such as \[1].
>
> * **RM-Baseline-2**: A REINFORCE-based method that uses a general-purpose pretrained reward model (Skywork-Reward-LLaMA-3.1-8B-v0.2) to compute delayed scalar rewards.
>
> * **RL-Baseline-1**: A variant inspired by GRPO, which samples multiple trajectories and computes an advantage estimate to replace the raw reward signal during policy optimization.
>
> * **RL-Baseline-2**: A baseline based on REINFORCE++ \[2], which employs more stable advantage estimation for improved gradient signals.
>
>     |Model||Reinforcement Learning||
>     |-|:-:|:-:|:-:|
>     ||MATH500|GPQA|Countdown|
>     |RM-Baseline-1|35.5(+1.1)|31.6(+1.3)|20.2(+6.1)|
>     |RM-Baseline-2|33.8(-0.6)|27.1(-3.2)|13.1(-1.0)|
>     |RL-Baseline-1|35.6(+1.2)|32.1(+1.8)|21.4(+7.3)|
>     |RL-Baseline-2|36.4(+2.0)|31.7(+1.4)|22.7(+8.6)|
>     |LLaDA + MRO|**37.4(+3.0)**|**33.8(+3.5)**|**27.2(+13.1)**|
>
> Due to the limited time available during the rebuttal phase, we focused on reproducing a representative subset of widely used RL baselines. Nevertheless, we believe these additional experiments are sufficient to demonstrate that our proposed **reward design consistently outperforms** not only d1-LLaDA but also a broader range of RL approaches and reward strategies. We will include these results and detailed discussions in the revised version for completeness.
>
> >*The method introduces significant computational overhead, especially for the token verification reward, which requires repeated model evaluations. Authors' claim that this is only a training-time cost does not address the fundamental scalability issue, especially as model and sequence sizes grow.*
>
> To further address the concern regarding training-time scalability, we conducted additional experiments using 1K RL training samples under varying generation lengths: 64, 128, 256, 512, 1024, and 2048 tokens. The wall-clock training times are summarized in the table below:
>
> | **Sequence Length** | **Total Time (h)** | **Relative Overhead vs. No-TV Reward** |
> | :-----------------: | :----------------: | :------------------------------------: |
> |         128         |        0.24        |                  1.14×                 |
> |         256         |        0.29        |                  1.16×                 |
> |         512         |        0.40        |                  1.13×                 |
> |         1024        |        0.58        |                  1.12×                 |
> |         2048        |        1.38        |                  1.15×                 |
>
> As shown above, although the total training time increases with sequence length, the **relative overhead introduced by the TV reward remains consistently modest**, even for sequences as long as 2048 tokens (only 1.15× compared to a no-TV baseline).
>
> This non-linear scaling is primarily due to two key optimizations:
>
> * **Using SGRO:** Instead of computing the TV reward at every denoising step, we sub-sample a fixed number of steps during training, significantly reducing computational cost while preserving effective training signals.
>
> * **Using Batch-parallelized Masked Evaluation:** Our implementation jointly batches all masked tokens for each denoising step, allowing for a single forward pass to evaluate multiple token positions in parallel. For example, for a denoising sequence like "ABC", we construct a batch such as \[\_BC, A\_C, AB\_] to efficiently evaluate the correlation of each token with just one pass through the model.
>
> Together, these optimizations ensure that **the TV reward remains computationally practical and scalable**, even as model size and sequence length grow. We promise to clarify these implementation details and runtime behaviors in the revised version.
>
>
> Reference:
> [1] Deepseekmath: Pushing the limits of mathematical reasoning in open language models
> [2] Reinforce++: A simple and efficient approach for aligning large language models
>
> ---
> We sincerely thank you for your positive feedback on our paper!
>
> Best,
>
> Authors

---

> > ### Author Response · Authors · 2025-08-06
> > **Reponse to Reviewer Sm7r**
> >
> > Dear Reviewer sm7r,
> >
> > Thank you once again for your thoughtful feedback and valuable suggestions. We apologize for reaching out again as the rebuttal deadline approaches, but we would like to highlight a few key additions we made in response to your concerns:
> >
> > * We provided a theoretical justification for one of the core component (i.e., TV reward) based on its connection to pairwise mutual information. This analysis shows that TV reward approximates intra-sequence correlation, offering a stronger conceptual foundation for its effectiveness.
> >
> > * We introduced a broader set of reward function baselines and RL method baselines beyond d1-LLaDA. We believe these additional comparisons demonstrate that our proposed reward design consistently outperforms not only d1-LLaDA, but also a more diverse range of RL approaches and reward strategies.
> >
> > * We conducted an empirical analysis of training-time overhead under varying generation lengths {64, 128, 256, 512, 1024, 2048}. Results show that although total training time increases with sequence length, the relative overhead introduced by TV reward remains consistently modest, even at 2048 tokens (only 1.15× compared to a no-TV baseline).
> >
> > We hope these clarifications and results have sufficiently addressed your concerns. If so, we would sincerely appreciate it if you could acknowledge our rebuttal and kindly consider updating your score.
> >
> > Thank you again for your time and engagement.
> >
> > Best regards,
> > The Authors

---

> ### Author Response · Authors · 2025-08-08
> **Reponse to Reviewer Sm7r**
>
> Dear Reviewer Sm7r,
>
> As the discussion period draws to a close, we would like to follow up once more to check if there are any remaining concerns or questions we can help clarify. We greatly value your feedback and have made a concerted effort to address each of your comments in detail.
>
> We noticed that the current rating remains negative. If you feel that our responses have adequately addressed your concerns, we would be sincerely grateful if you could consider updating your score. Please feel free to let us know if there is anything that still requires clarification.
>
> Thank you again for your time and thoughtful engagement.
>
> Best regards,
> The Authors

---

> ### Author Response · Authors · 2025-08-09
>
> Dear Reviewer Sm7r,
>
> As the discussion period draws to a close, we would like to check whether our rebuttal has adequately addressed your concerns. If you have any remaining questions or points that require clarification, please don't hesitate to let us know. Thank you once again for your time and engagement.
>
> Best regards,
> The Authors

---

### Official Review · Reviewer_9xuo · 2025-06-29

**Clarity:** 2
**Significance:** 2
**Originality:** 3
**Rating:** 3
**Confidence:** 4

**Summary:**

The authors point out that diffusion language models face the problem of missing intra-sequence and inter-sequence correlations in reasoning tasks. They propose the Multi-Reward Optimization (MRO) framework, introducing three types of rewards—Token Verification, external language-model perplexity, and answer-level reasoning correctness—and present the Step-wise Group Reward Optimization method to reduce variance. Experiments conducted in the three modes of test-time scaling, rejection sampling, and reinforcement learning confirm the effectiveness of the proposed approach.

**Questions:**

1. Section 5.3 (“Test-time Scaling”) is puzzling. Lines 304–305 say that policy-gradient updates are carried out during “test time.” If back-propagation is involved, can this still be called test-time scaling? Moreover, a detailed explanation of how the policy gradients are computed would greatly aid reader understanding.

2. When evaluating the LLaDA-Instruct baseline, did you enable semi-autoregressive sampling? If so, what block length did you use?

**Ethical Concerns:**

["NO or VERY MINOR ethics concerns only"]

**Final Justification:**

I believe my concern was not adequately addressed in the rebuttal. As detailed in Replying to Rebuttal by Authors, issues such as imprecise mathematical notation and potentially misleading writing still remain. Therefore, I am keeping my score as Borderline Reject.

**Limitations:**

Yes

**Quality:**

2

**Strengths And Weaknesses:**

## Strengths
1. The paper is the first to investigate how intra-sequence correlation affects reasoning performance in diffusion language models—a crucial issue, especially when the number of sampling steps is small.

## Weakness
1. The mathematical definition of the x-axis in Figure 1 is never specified.

2. Equation (3) appears incorrect. The left-hand side of the first equality is the marginal distribution of $r_0$, whereas the right-hand side is the joint distribution of $r_0, r_1, \dots, r_{t-1}$. The second equality is also wrong; please refer to Eq. (7) in [1] for the standard backward-transition formula for masked diffusion models.

3. Token Verification Reward lacks theoretical grounding. The paper introduces this reward but does not explain why it should faithfully measure—or improve—intra-sequence correlation.

4. The authors treat the final-answer correctness (Eq. 6) as an inter-sequence reward, yet final-answer accuracy also depends on intra-sequence correlation.

5. Computational overhead is unreported. The comparison between Figure 2 and Figure 5 is not rigorous because MRO introduces substantial extra computation. The authors should quantify the additional wall-clock time or FLOPs required.

[1] Sahoo et al. Simple and Effective Masked Diffusion Language Models. NeurIPS 2024.

---

> ### Author Rebuttal · Authors · 2025-07-31
>
> Dear Reviewer 9xuo,
>
> We appreciate your recognition that "our paper is the first to investigate how intra-sequence correlation affects reasoning performance in diffusion language models", and that this investigated issue is "crucial".
>
> We will provide explanations for the main points that you are concerned about.
>
> ---
>
> >*W1: The mathematical definition of the x-axis in Figure 1 is not specified.*
>
> Thanks for your helpful feedback! The x-axis in Figure 1 represents the intra- and inter-sequence correlation rewards defined in Section 4.2: the left part shows the cumulative intra-sequence rewards $R^{tv}\_{t} + R^{ppl}\_{t}$, and the right part shows the inter-sequence reward $R^{q}_{0}$. As the reward scales differ, we apply standardization for visualization. We will add this clarification to the introduction for better accessibility.
>
> >*W2: In Equation 3, it is unclear why the left-hand side of the first equality represents the marginal distribution of $r_0$, while the right-hand side corresponds to the joint distribution of $\{r_0, r_1, \dots, r_{T-1}\}$. The same issue applies to the second equality.*
>
> We apologize for the confusion caused by Equation 3. Below, we clarify the roles of the two distributions presented in the equation.
>
> - Left-hand side:
>    The symbol $\mathrm{Pr}'\_{\theta}(r_{0}\mid p_{0},r_{T})$ is the decoding distribution, i.e., the marginal distribution over the final response $r_{0}$ after $T$ denoising steps.  This is the quantity we actually sample at inference time.
>
> - Right-hand side:
>    The product
>
>    $\prod\_{t=T}^{1}\mathrm{Pr}\_{\theta}(r_{t-1}\mid p_{0},r_{t})$
>
>    is the joint denoising distribution over the entire latent sequence $\{r_{T},r_{T-1},\dots,r_{0}\}$.
>    - Each factor $\mathrm{Pr}\_{\theta}(r_{t-1}\mid p_{0},r_{t})$ conditions only on the current masked sequence $r_{t}$ and predicts the corresponding masked tokens.
>    - Marginalizing the latent variables $\{r_{1},\dots,r_{T-1}\}$ yields the decoding distribution on the left-hand side.
>
> - Second equality:
>    The further factorization
>
>    $\prod\_{i=1}^{L_{r}}\mathbf{1}[r_{t-1}^{i}=M]\,\mathrm{Pr}\_{\theta}(r_{t-1}^{i}\mid p_{0},r_{t})$
>
>    explicitly reflects that, within each denoising step, masked tokens are predicted in parallel and independently, conditioned on the current input $r_t$. The indicator function $\mathbf{1}[r_{t-1}^i = M]$ ensures that only masked tokens are predicted.
>
> In essence, Equation 3 expresses that the decoding distribution over \$r\_0\$ is constructed through a chain of \$T\$ denoising steps, where each step factorizes over masked tokens. This formulation is equivalent to that used in Sahoo et al. (2024), with the key difference that we employ an indicator function instead of an explicit conditional to present the token-wise structure more concisely. A similar formulation also appears in the original LLaDA paper, which we adopt here for consistency.
>
> We promise to add this detailed explanation and revise the presentation in the revision to prevent potential confusion.
>
> >*W3: Token Verification Reward (TVR) lacks theoretical grounding.*
>
> We provide a theoretical proof for the TVR by analyzing it from the perspective of mutual information, showing that it effectively promotes intra-sequence correlation.
>
> Recall the definition of the TVR at the denoising step $t$:
>
> $
> R^{\text{tv}}\_{t}= \frac{1}{N}\sum\_{n=1}^{N}\Pr\_{\theta}(r\_{t}^{m\_{n}}\mid p\_{0},r\_{t-1}/\{r\_{t-1}^{m\_{n}}\}),
> $
>
> where $M = \{m_1, \dots, m_N\}$ is the set of masked token indices, $r_{t-1}^{M}$ is the set of the predicted tokens at these positions, and $r_{t-1} / r_{t-1}^{M}$ is the set of unmasked token set. The joint probability over the masked positions can be factorized autoregressively (within the masked set) as:
>
> $
> \Pr_{\theta}(r_{t-1}^{M} \mid p_0, r_t)
> = \prod_{n=1}^{N} \Pr_{\theta}(r_{t-1}^{m_n} \mid p_0, r_{t-1} / r_{t-1}^{M}, r_{t-1}^{<m_n}),
> $
>
> where the rest of the sequence is fixed and only the masked tokens are predicted.
>
> TVR approximates this joint modeling by computing **leave-one-out log-probabilities**:
>
> $
> R^{\text{tv}}\_{t}= \frac{1}{N} \sum_{n=1}^{N} \log \Pr_{\theta}(r_{t}^{m_n} \mid p_0, r_{t-1} / r_{t-1}^{m_n})
> $
>
> Now define the empirical average pairwise mutual information (PMI) among the masked tokens:
>
> $
> \mathrm{PMI}\_{\text{avg}} = \frac{2}{N(N-1)} \sum_{1 \le i < j \le N} \mathrm{I}(r_{t-1}^{m_i}; r_{t-1}^{m_j} \mid p_0, r_{t-1} / r_{t-1}^{M})
> $
>
> Using a standard second-order Taylor expansion around the independence assumption (as in energy-based models), this quantity can be approximated as:
>
> $
> \mathrm{PMI}\_{\text{avg}} \approx \frac{2}{N(N-1)} \sum_{i < j} \log \frac{\Pr_{\theta}(r_{t-1}^{m_i}, r_{t-1}^{m_j} \mid \cdot)}{\Pr_{\theta}(r_{t-1}^{m_i} \mid \cdot) \Pr_{\theta}(r_{t-1}^{m_j} \mid \cdot)}
> $
>
> Importantly, the leave-one-out log-probabilities computed in TVR serve as sufficient statistics for estimating these pairwise interactions. Therefore, **maximizing the average leave-one-out log-probability is first-order equivalent to maximizing $\mathrm{PMI}_{\text{avg}}$, thereby promoting stronger intra-sequence correlation**. We promise to add the theoretical grounding in the revision.
>
> >*W4: The authors treat the final-answer correctness (Eq. 6) as an inter-sequence reward, yet final-answer accuracy also depends on intra-sequence correlation.*
>
> Thanks for your insightful comment! We would like to clarify that **our reward categorization is intended to support the modeling and optimization of token correlations in DLMs, rather than to define independent reward signals**. Specifically, the intra-sequence rewards are designed to capture token-level correlations within a single sequence. In contrast, the inter-sequence reward is only observable after the entire denoising process is complete and is intended to assess the correlation among sequences generated at different denoising steps, i.e., whether they collectively produce an accurate final answer.
>
> >*W5: Computational overhead is unreported in Figure 2 and Figure 5.*
>
> Below, we provide clarifications regarding the computational overhead reported in Figures 2 and 5. **Our results demonstrate that MRO is computationally efficient and well-suited for real-world deployment**.
>
> 1. For the test-time scaling (TTS) experiments in Figure 2, we provide a detailed latency comparison between vanilla decoding and MRO, as shown in the table below. Given that TTS requires additional sampling and thus introduces time overhead, we also include a comparison with a confidence-based reward TTS baseline, which involves the same sampling cost as our MRO but does not require extra reward computation. From the results, we see that **while reward computation does introduce some overhead, it remains within a reasonable range and is justified by the substantial performance gains it brings**. Compared to the confidence-based reward TTS baseline, we further see that **when excluding the time cost introduced by sampling, our reward design incurs only minimal additional overhead while delivering substantial performance gains**.
>
>     |Method|MATH500|  |GPQA|  |
>     |-|:-|:-:|:-|:-:|
>     ||Decoding Time |Score|Decoding Time|Score|
>     |Vanilla (LLaDA)| 0.35h~0.39h |33.2 | 0.16h~0.21h |29.2|
>     |LLaDA-TTS + Confidence-based Reward| 0.73h~0.81h |35.2 | 0.27h~0.34h |30.6|
>     |LLaDA-TTS + MRO| 0.84h~0.85h |36.0 | 0.31h~0.44h |34.6|
>
> 2. **In Figure 5, we would like to clarify that in both the rejection sampling and RL fine-tuning settings, multi-reward computation and policy gradient updates are performed only during training**. Once the model parameters are updated, decoding proceeds identically to the standard LLaDA [22], with no additional reward computation or forward passes involved, and thus, the end-to-end decoding latency remains unaffected. This is further demonstrated in the table below, where the comparison confirms that our rejection sampling and RL setups do not introduce noticeable decoding-time latency on eight A800 GPUs.
>
>     |Method|MATH500 |  |GPQA|  |
>     |-|:-|:-:|:-|:-:|
>     ||Decoding Time |Score|Decoding Time|Score|
>     |Vanilla Decoding (LLaDA)| 0.35h~0.39h |33.2 | 0.16h~0.21h |29.2 |
>     |LLaDA-RS + MRO| 0.34h~0.37h |34.2 | 0.14h~0.18h |32.1 |
>     |LLaDA-RL + MRO| 0.36h~0.39h |35.2 | 0.16h~0.20h |33.8 |
>
> >*Q1: Section 5.3 ("Test-time Scaling") is puzzling. Lines 304–305 say that policy-gradient updates are carried out during "test time". If back-propagation is involved, can this still be called test-time scaling?*
>
> We respectfully clarify that **policy gradient updates occur only during training, not at test time**. The TTS procedure serves purely as a diagnostic tool to evaluate reward signal quality without any backpropagation or parameter updates, as also noted in line 273. For clarity, we will move this explanation to a more prominent location. Indeed, our approach aligns with OpenAI's RLHF evaluation protocol \[39], where reward models are first assessed via TTS before guiding methods like rejection sampling or RL. **Lines 304–305 aim to highlight that our TTS analysis validates the reward signal’s effectiveness for downstream policy optimization.**
>
>
> >*Q2: When evaluating the LLaDA-Instruct, did you enable semi-autoregressive sampling? If so, what block length did you use?*
>
> Yes, we follow the official evaluation code released by LLaDA, which includes semi-autoregressive sampling by default. The block length is set according to the recommended configuration in the original LLaDA paper, as shown below.
>
> ||GSM8K|MATH500|GPQA|Countdown|Sudoku|
> |-|:-:|:-:|:-:|:-:|:-:|
> |Block Length|8|64|64|64|64
>
> ---
> We truly appreciate your positive feedback on our paper!
>
> Best,
>
> Authors

---

> > ### Comment · Reviewer_9xuo · 2025-08-04
> >
> > Thank you for the response. I find that some parts of the rebuttal contain factual inaccuracies and do not fully address my concerns.
> >
> > In light of W2, I find the mathematical notation to be imprecise. The authors explained that $Pr_{\theta}(r_0|p_0, r_T)$ denotes the marginal distribution of $r_0$, while $\prod_{t=T}^1 Pr_{\theta}(r_{t-1}|p_0, r_t)$ represents the joint distribution of $r_T, r_{T-1}, \dots, r_0$. My concern is that these two distributions cannot be connected by an equality sign. A correct expression should be something like $Pr_{\theta}(r0|p_0, r_T) =\int \prod_{t=T}^1 Pr_{\theta}(r_{t-1}|p_0, r_t) dr_{1,\dots,T}$. After reviewing references [1] and [2], I did not find any instances where a marginal distribution is equated to a joint distribution.
> >
> > Regarding W4, I find the authors’ categorization of the rewards to be confusing and imprecise. In the submission, Section 4.2.1 is titled Intra-sequence Correlation Rewards and Section 4.2.2 is titled Inter-sequence Correlation Rewards, which clearly leads readers to assume that these rewards are designed to address different types of correlation errors (i.e., intra- vs. inter-sequence correlation). However, in the rebuttal, the authors attempt to deny this implication without modifying the wording in the paper, which I find unconvincing.
> >
> > [1] Sahoo et al. Simple and Effective Masked Diffusion Language Models.
> >
> > [2] Nie et al. Large Language Diffusion Models.

---

> > > ### Author Response · Authors · 2025-08-04
> > > **Reponse to Reviewer 9xuo**
> > >
> > > Dear Reviewer 9xuo,
> > >
> > > Thank you again for your thoughtful feedback and for continuing to engage with our paper. Below we address your remaining concern regarding W2 and W4.
> > >
> > > ----
> > >
> > > >*W2: In light of W2, I find the mathematical notation to be imprecise. ... My concern is that these two distributions cannot be connected by an equality sign.*
> > >
> > >
> > > We apologize for not fully resolving this issue in our initial response. We would like to clarify that in the first equality, the symbol $\mathrm{Pr}'\_{\theta}(r_{0} \mid p_{0}, r_{T})$ (please note the prime symbol) refers to the **decoding distribution**, while $\mathrm{Pr}\_{\theta}(r_{t-1} \mid p_{0}, r_t)$ denotes the **denoising distribution** learned by the DLM.
> > >
> > > In practice, during DLM inference, we construct the decoding distribution $\mathrm{Pr}'$ by **chaining together the denoising distributions** $\mathrm{Pr}\_{\theta}(r_{t-1} \mid p_0, r_t)$ through a sequence of denoising steps starting from $r_T$. Thus, the decoding distribution is **not a marginal of the denoising joint distribution**, but rather a composition of multiple conditionals, as is standard in diffusion-based generation. **This same formulation appears in Equation 6 of Xu et al. (2025) \[1], which also models the decoding distribution as a sequential composition of learned denoising steps**.
> > >
> > > Thanks for your helpful feedback, and we promise to revise the notation in the final version to make this distinction between decoding and denoising distributions explicit and mathematically precise.
> > >
> > >
> > > >*W4: Regarding W4, I find the authors’ categorization of the rewards to be confusing and imprecise. ... However, in the rebuttal, the authors attempt to deny this implication without modifying the wording in the paper, which I find unconvincing.*
> > >
> > > We sincerely apologize for the confusion and appreciate your close reading. We would like to clarify that our statement in the rebuttal, *“our reward categorization is intended to support the modeling and optimization of token correlations in DLMs, rather than to define independent reward signals”*, is **not intended to deny the distinction implied by our section headings or reward naming**.
> > >
> > > Instead, we intend to respond specifically to your earlier point that *“final-answer accuracy also depends on intra-sequence correlation”*. Our clarification is meant to emphasize that while the intra- and inter-sequence rewards are conceptually distinct in **design and optimization intent**, they are **not completely disentangled in effect**, in other words, they may both be influenced by overlapping factors such as token-level dependencies.
> > >
> > > To make this more straightforward, consider a simplified example: suppose we define two rewards: $R^{\mathrm{tv}}$ for intra-sequence correlation and $R^{\mathrm{q}}$ for inter-sequence correlation. Both rewards may vary with respect to a shared latent factor (e.g., token dependency strength), but they are still **targeting different aspects of model behavior**. Specifically, $R_1$ focuses on local consistency within a single sequence, while $R_2$ evaluates global coherence across denoising steps. Thus, although the signals may interact, their objectives and roles in training are meaningfully different.
> > >
> > > This distinction is further supported by the ablation study presented in Appendix C.3, where we isolate the effects of the two reward types. Specifically, we evaluate two variants: MRO-v1, which uses only intra-sequence rewards for training, and MRO-v3, which uses only inter-sequence rewards. On the MATH500 benchmark under the test-time scaling protocol, MRO-v1 achieves a score of 36.8, and MRO-v3 achieves 35.6. In contrast, when both rewards are optimized jointly in the full MRO model, the performance improves to 38.0. **These results confirm that while the rewards are not completely independent, they target complementary training objectives within our framework**.
> > >
> > > We appreciate this opportunity to clarify, and promise to revise the reward descriptions in the final version to make this distinction more explicit and avoid potential confusion.
> > >
> > > Reference:
> > > [1] Xu et al. Energy-Based Diffusion Language Models for Text Generation. ICLR 2025.
> > >
> > >
> > > ---
> > >
> > > We sincerely appreciate your positive feedback and thank you again for your time.
> > >
> > > Best,
> > >
> > > Authors

---

> > > > ### Comment · Reviewer_9xuo · 2025-08-05
> > > >
> > > > The authors appear to have misunderstood a critical issue I raised. In the rebuttal, they incorrectly equated the distribution of a single random variable, $p(r_0)$, with the joint distribution over multiple variables, $p(r_0, r_1, \dots, r_T)$. I had already pointed out the correct formulation in my initial comments (i.e., $p(r_0)=\int p(r_0, r_1, \dots, r_T) d r_1, \dots, r_T$). I strongly encourage the authors to carefully compare the equation in the submission with what they claim to be “the same formulation”  (i.e., Equation 6 in [1]) and reflect on the differences.
> > > >
> > > >
> > > > In addition, the paper explicitly categorizes reward signals as Intra-sequence Correlation and Inter-sequence Correlation, placing the final-answer reward under the latter. However, based on both my understanding and the authors’ rebuttal, the final-answer reward in fact optimizes both intra- and inter-sequence aspects. This inconsistency is confusing, and the rebuttal did not attempt to resolve it. Instead, it confirms that my concern was well-founded.
> > > >
> > > > [1] Xu et al. Energy-Based Diffusion Language Models for Text Generation. ICLR 2025.

---

> ### Author Response · Authors · 2025-08-05
> **Reponse to Reviewer 9xuo**
>
> Dear Reviewer 9xuo,
>
>
> Thank you once again for your thoughtful response and continued engagement. Please allow me to offer a further clarification regarding your concern.
>
> ---
> >*The authors appear to have misunderstood a critical issue I raised. ... I strongly encourage the authors to carefully compare the equation in the submission with what they claim to be “the same formulation” (i.e., Equation 6 in [1]) and reflect on the differences.*
>
> Thank you for your continued feedback. With this discussion, we now better recognize the source of the confusion and would like to offer a more precise clarification.
>
> We fully agree that the expression you suggested,
>
> $
> p(r_0) = \int p(r_0, r_1, \dots, r_T) \, d(r_1, \dots, r_T)
> $
>
> is mathematically sound and correctly defines the decoding distribution as a marginalization over the full joint distribution. This formulation provides a rigorous foundation and allows for an equality to hold.
>
> However, in practice, such integrals over the full denoising trajectory are computationally intractable during decoding. Instead, DLMs approximate the decoding distribution through a sequential denoising process:
>
> $
> \prod_{t=T}^{1} p(r_{t-1} \mid r_t)
> $
>
> This expression reflects the fact that generating $r_0$ requires iteratively denoising intermediate sequences $\{r_{T-1}, r_{T-2}, \dots, r_1\}$, and is a commonly adopted approach in practice.
>
> That said, we now understand that the first equation in our original submission may have been misleading. Although we introduced a separate symbol $\mathrm{Pr}'(\cdot)$ to distinguish the decoding distribution from the learned denoising distributions $\mathrm{Pr}(\cdot)$, this notation was insufficient to clearly communicate the relationship between the marginal and the compositional approximation.
>
> To address this, we will revise the formulation more explicitly in the final version. Specifically, we *define*:
>
> $
> q(r_{T-1}, \dots, r_0 \mid  r_T) = \prod_{t=T}^{1} p(r_{t-1} \mid r_t)
> $
>
> where $q(\cdot)$ denotes the distribution over the entire denoising trajectory leading to $r_0$, and $p(\cdot)$ corresponds to the denoising distribution at each time step.
>
> We truly appreciate your suggestion to view decoding from an integral-based perspective—it provides a more complete mathematical grounding. We will explore such a formulation in future work, and we believe it could lead to promising directions in improving the theoretical rigor and practical efficiency of DLM decoding.
>
> We promise to revise the text accordingly to clearly differentiate between the marginal decoding distribution and its sequential approximation, based on this discussion.
>
> >*In addition, the paper explicitly categorizes reward signals as Intra-sequence Correlation and Inter-sequence Correlation, placing the final-answer reward under the latter. However, based on both my understanding and the authors’ rebuttal, the final-answer reward in fact optimizes both intra- and inter-sequence aspects. This inconsistency is confusing, and the rebuttal did not attempt to resolve it. Instead, it confirms that my concern was well-founded.*
>
>
> Through this discussion, we recognize that the current naming may have caused confusion, and we appreciate your feedback. In the revised version, we will reorganize the section headings and explicitly clarify the relationship between the different rewards, as well as their distinct optimization objectives, based on the insights from this discussion.
>
>
> ---
>
> We sincerely appreciate your positive feedback and thank you again for your time.
>
> Best,
>
> Authors

---

> > ### Author Response · Authors · 2025-08-06
> > **Reponse to Reviewer 9xuo**
> >
> > Dear Reviewer 9xuo,
> >
> > We apologize for reaching out again as the discussion deadline approaches, and we sincerely appreciate the comprehensive and constructive feedback you have provided throughout this process.
> >
> > We are glad that the new results we provided have addressed most of your concerns. However, due to the limitations of the NeurIPS rebuttal process, we were unable to update our paper during the discussion phase. As a result, we could not revise the formulation of the token correlation problem or improve the clarity of the reward definitions in the current version.
> >
> > We promise to assure you that we will incorporate your suggestions in the revised version. Specifically:
> >
> > * We will revise Equation (3) by using distinct notations to clearly differentiate between $\mathrm{Pr}'$ and $\mathrm{Pr}$. Additionally, we will explicitly express the denoising sequence distribution on the left-hand side of the equation using $q(r_{T-1}, \dots, r_0 \mid r_T)$ to ensure full mathematical rigor.
> >
> > * We will reorganize the section structure and clarify the relationship between the different reward components, as well as their distinct optimization roles, based on the insights gained through our discussion.
> >
> > We hope these planned revisions adequately address your remaining concerns. If so, we would be sincerely grateful if you could acknowledge our rebuttal and kindly consider updating your score.
> >
> > Thank you once again for your time and thoughtful engagement.
> >
> > Best regards,
> > The Authors

---

> > ### Comment · Reviewer_9xuo · 2025-08-07
> >
> > Thank you for the response.
> >
> > Regarding the first issue, I believe the authors have a misunderstanding of the formulation of diffusion models. Integrating over the joint distribution of data and latent variables to obtain the data distribution does not mean that this integral must be computed analytically during sampling. I recommend that the authors refer to Section 2 of [1] for a better understanding of this concept.
> >
> > For the second issue, my concern is that the structure of the paper could lead to confusion and misunderstanding for readers. The authors have acknowledged this point. Although they promised to revise it in the final version, I believe such a revision would fundamentally change the structure and claims of the paper.
> >
> > Taking everything into consideration, I have decided to keep my score.
> >
> > [1] Song et al. Denoising diffusion implicit models.

---

> > > ### Author Response · Authors · 2025-08-08
> > > **Reponse to Reviewer 9xuo**
> > >
> > > Dear Reviewer 9xuo,
> > >
> > > Thank you for taking the time to provide such thoughtful and constructive feedback throughout the discussion phase.
> > >
> > > While we regret your decision to keep your score, we are sincerely grateful for your continued engagement.
> > >
> > > As promised, we are fully committed to addressing these concerns in the final version. Specifically, we will revise the mathematical formulation to more closely align with standard conventions in diffusion modeling, and we will reorganize the reward-related sections to avoid structural ambiguity. To clarify, these revisions will focus on improving the clarity and presentation of our method rather than altering its core technical contributions or experimental findings. In particular, we will explicitly highlight how our reward components are designed to optimize intra- and inter-sequence correlations, rather than portraying them as strictly isolated categories (i.e., “intra-sequence” vs. “inter-sequence” rewards).
> > >
> > > We want to reassure you that the central contribution of our work—optimizing token correlations in DLMs through multi-reward training—remains intact and unaffected by these adjustments.
> > >
> > > Thank you again for your valuable feedback and for helping us improve the clarity and rigor of our paper.
> > >
> > > Best regards,
> > > The Authors

---

### Official Review · Reviewer_CEHF · 2025-07-01

**Clarity:** 3
**Significance:** 2
**Originality:** 2
**Rating:** 5
**Confidence:** 2

**Summary:**

This paper proposes Multi-Reward Optimization (MRO), a novel framework designed to enhance the reasoning capabilities of Diffusion Language Models (DLMs). The key motivation is that the parallel token generation process in DLMs lacks token correlation, which hampers performance on reasoning tasks. To address this, the authors introduce intra- and inter-sequence correlation rewards and optimize them using test-time scaling, rejection sampling, and reinforcement learning.

**Questions:**

See weakness above.

**Ethical Concerns:**

["NO or VERY MINOR ethics concerns only"]

**Final Justification:**

I decide to keep my score.

**Limitations:**

Yes.

**Paper Formatting Concerns:**

No.

**Quality:**

3

**Strengths And Weaknesses:**

Pros:
1. The paper studied an important problem. The authors address a key limitation of DLMs, namely the lack of token correlation, by designing multiple reward signals tailored to different aspects of reasoning.
2. Extensive experiments across five reasoning benchmarks (e.g., GSM8K, MATH500, GPQA) demonstrate that MRO substantially improves performance and sample efficiency.
3. The definition of the problem is clear. The authors pinpoint token-correlation deficiencies as a root cause of poor DLM reasoning.

Cons:
1. The reliance on repeated forward passes (especially in test-time scaling and rejection sampling) increases inference and training overhead. Can the authors comment on this?
2. Can the authors also report the results on long-reasoning benchmarks such as AIME and AMC? This will help the readers to better judge the effectiveness of the proposed method and build future works on top of it.

---

> ### Author Rebuttal · Authors · 2025-07-31
>
> Dear Reviewer CEHF,
>
> We appreciate the reviewer’s constructive and thoughtful feedback.
>
> Thanks for your recognition that our paper "studied an important problem" and clearly defines the core issue by "pinpointing token-correlation deficiencies as a root cause of poor DLM reasoning". We are also grateful for your findings that our MRO addresses this limitation by "designing multiple reward signals tailored to different aspects of reasoning", and that our "extensive experiments across five reasoning benchmarks" demonstrate that MRO "substantially improves performance and sample efficiency".
>
> We provide explanations for the main points that you are concerned about below.
>
> ---
> >*W1: The reliance on repeated forward passes, especially in test-time scaling (TTS) and rejection sampling, increases inference and training overhead. Can the authors comment on this?*
>
> Thanks for your insightful comment! Below, we clarify the issue related to inference/training time and present empirical results to support our claims. **Taken together, we demonstrate that our MRO is highly practical and well-suited for real-world applications**.
>
> - Regarding your concern about inference-time overhead, we first would like to clarify that in both the rejection sampling and RL fine-tuning settings, multi-reward computation and policy gradient updates are performed **only during training**. Once the model parameters are updated, decoding proceeds identically to the standard LLaDA baseline \[22], with **no additional reward computation or repeated forward passes** involved during inference.
>
> - Additionally, to further address your concern about the overhead introduced by our reward computation, we provide a detailed breakdown of the associated computational cost in the table below, where $\Theta$ denotes the model size and $L$ denotes the sequence length. To reduce the overall burden, we adopt several efficiency-oriented strategies. Specifically, **we leverage SGRO to sparsely compute rewards across decoding steps, significantly limiting the number of evaluations needed**. For the Perplexity Reward, we employ a lightweight model (GPT-2-small) to minimize FLOPs. We also employ GPU-parallelized computation to process multiple reward evaluations concurrently during training, substantially reducing wall-clock latency.
>
>     |Reward Type| Model Used | Formula (Per Occurrence) | Occurrence Count (w/ SGRO) |Total FLOPs (Estimated)|
>     | - | :-: | :-: | :-: | :-: |
>     |Token Verification Reward| LLaDA-8B  | $2 \cdot \Theta_{\text{LLaDA}} \cdot L \cdot (N + 1)$ | $\frac{T}{w} = \frac{128}{w}$    | $\frac{128}{w} \cdot 1.23 \times 10^{13}$ |
>     |Perplexity Reward| GPT-2-Small  | $2 \cdot \Theta_{\text{GPT-2}} \cdot L$ | $\frac{T}{w} = \frac{128}{w}$    | $\frac{128}{w} \cdot 6.0 \times 10^{10}$  |
>
>
> - We have also conducted a comparison of both inference and training performance with and without our reward design, as shown in the tables below. The runtime is measured on eight A800 GPUs with a length of 256 under data parallelism, with each configuration run five times to report a consistent range of values. The results demonstrate that **our reward introduces only a modest computational overhead while delivering substantial performance improvements**, confirming its practicality and effectiveness.
>
>     |Method|Inference Time (MATH500)|Training Time|Score (MATH500)|
>     |-|:-:|:-:|:-:|
>     |LLaDA|0.35h~0.39h|-|33.2|
>     ||
>     |LLaDA-RS w/o MRO|0.35h~0.38h|10.3h~11.0h|33.0|
>     |LLaDA-RS w/ MRO|0.34h~0.37h|12.7h~13.1h|34.2|
>     ||
>     |LLaDA-RL w/o MRO|0.33h~0.37h|8.2h~8.4h|33.4|
>     |LLaDA-RL w/ MRO|0.36h~0.39h|9.6h~10.1h|35.2|
>     ||
>     |LLaDA-TTS w/o MRO|0.73h~0.81h|-|35.2|
>     |LLaDA-TTS w/ MRO|0.84h~0.85h|-|36.0|
>
>
> >*W2: Can the authors also report the results on long-reasoning benchmarks such as AIME and AMC? This will help the readers to better judge the effectiveness of the proposed method and build future works on top of it.*
>
>
> |Model|AIME2024|AMC2023|
> |-|:-:|:-:|
> |LLaMA-3-8B-Instruction|6.7|30.0|
> |Qwen2.5-7B-Instruct|10.0|52.5|
> ||
> |LLaDA|3.3|17.5|
> |LLaDA-TTS + MRO|6.7|22.5|
> |LLaDA-RS + MRO|**10.0**|**25.0**|
> |LLaDA-RL + MRO|6.7|20.0|
>
> - The results show that our proposed MRO remains effective on long-form reasoning tasks, consistently enhancing the reasoning capabilities of LLaDA even at a length of 768. Notably, MRO helps close the performance gap between DLMs and AR-style models. These findings further support that MRO can effectively address a fundamental weakness of DLMs, the lack of intra-/inter-sequence correlation modeling.
>
> - We observe that DLMs continue to underperform on long-form reasoning tasks, which we attribute to their relatively early stage of development compared to AR-based models. This observation underscores the significance of our work: MRO directly tackles one of the key challenges in DLM reasoning (i.e., the lack of token-level correlation modeling) and represents a meaningful step toward enhancing the reasoning capabilities of this emerging class of models.
>
> ---
> We sincerely appreciate your positive feedback and thank you again for your time.
>
> Best,
>
> Authors

---

> > ### Comment · Reviewer_CEHF · 2025-08-03
> > **Reponse to authors**
> >
> > Thank the authors for the response. I decide to maintain my rating.

---

> > > ### Author Response · Authors · 2025-08-04
> > >
> > > We truly appreciate your response. Should you have any further questions or suggestions, we would be more than happy to provide additional clarification or discuss them further.

---

### Official Review · Reviewer_9WDq · 2025-07-02

**Clarity:** 2
**Significance:** 3
**Originality:** 2
**Rating:** 4
**Confidence:** 4

**Summary:**

This paper proposes a novel approach to improve the reasoning capabilities of diffusion language models (DLMs) by optimizing token correlations through a Multi-Reward Optimization (MRO) framework.

The key contributions of this paper are as follows:

1.	Intra-sequence and Inter-sequence correlation: The authors define two types of token correlations: intra-sequence correlation and inter-sequence correlation, to capture token correlations across denoising steps.

2.	Multi-Reward Optimization (MRO) framework: The authors propose an MRO approach that enables DLMs to generate reasoning path with an emphasis on token correlation.

3.	Step-wise Group Reward Optimization (SGRO): The authors introduce SGRO in the optimization phase to reduce reward variance, leading to more stable optimization.

4.	Comprehensive evaluation: Through extensive experiments on various reasoning tasks, the authors demonstrate that MRO significantly improves the reasoning performance of DLMs.

**Questions:**

1.	In Equation 5, the rationale for setting the upper bound of the perplexity reward to 100 is unclear. Have your test alternative parameters, and if so, how do they impact the RL training process?
2.	In Equation 6, considering only r_0 in the inter-sequence correlation rewards is questionable. In a Markov setting, this reward should be able to measure the correlation between consecutive state pairs (e.g., s_t and s_{t-1}), and all consecutive state pairs should be considered in a complete reasoning trajectory.
3.	In Equation 8, each separate reward has a different range, resulting in varying weights in the total reward. Why is reward normalization not utilized to address this problem?
1.	Typos and notation abuse:
(1)	Line 159: the response x_0 should be r_0
(2)	Line 168: p_{theta}(x_0|x_t) is for the pretraining stage, not the SFT stage
(3)	Line 253: the “<” in the variance inequality should be “>”
(4)	For the figure 7 in the appendix, the title and the x-axis label should be “group size”, not “temperature”

**Ethical Concerns:**

["NO or VERY MINOR ethics concerns only"]

**Final Justification:**

I appreciate the authors’ detailed responses, which effectively address my concerns. I have accordingly raised my score. I strongly recommend that the authors include the additional experiments and analyses in the final version of the paper.

**Limitations:**

YES

**Quality:**

2

**Strengths And Weaknesses:**

Strengths:

1.	Thorough analysis: The authors provide a thorough analysis of the limitations of current DLMs in reasoning tasks, specifically focusing on the lack of token correlation across denoising steps.

2.	Detailed framework explanations: The authors provide detailed explanations of the proposed MRO framework, including the design of intra-sequence, inter-sequence correlation rewards and SGRO.

3.	Comprehensive experiments: The paper includes extensive experiments on multiple reasoning tasks across different models and settings. The results consistently demonstrate the effectiveness of the proposed MRO framework.

Weaknesses

1.	Lack of clarity in the Introduction: The main contributions of the paper are not clearly articulated in the introduction.

2.	Inadequate figures and pseudocodes: The paper lacks a comprehensive figure illustrating the MRO framework workflow, which would help readers in understanding all components of MRO. Additionally, the inclusion of brief pseudocode for algorithms such as SGRO is recommended.

3.	Typos and notation abuse: There are some typos and inconsistent notations throughout the paper, which hinder clarity.

4.	Ambiguous reward settings: Some reward settings appear unreasonable; further details can be found in the Question section.

---

> ### Author Rebuttal · Authors · 2025-07-31
>
> Dear Reviewer 9WDq,
>
> We sincerely thank the reviewer for your positive and insightful feedback.
>
> We greatly appreciate your recognition of our paper’s "thorough analysis of the limitations of current DLMs in reasoning tasks", particularly the focus on "the lack of token correlation across denoising steps". We are also pleased that our "detailed explanations of the proposed MRO framework", including the design of intra-/inter-sequence correlation rewards, and SGRO, are found to be clear. Finally, we are grateful for your recognition that the paper presents "extensive experiments on multiple reasoning tasks across different models and settings".
>
> We will provide explanations for the main points that you are concerned about.
>
> ---
>
> > *W1: The main contributions of the paper can be clearly articulated in the introduction.*
>
> Thanks for your helpful suggestion! We intend to convey the core contributions throughout the introductory section. Specifically, as noted in line 73, we introduce MRO as an approach to enhance token correlation in DLMs without introducing additional architectural burdens. In line 81, we describe SGRO, our sparse reward optimization strategy, designed to further reduce computational overhead. Finally, in line 86, we summarize the empirical findings from extensive experiments across test-time scaling, rejection sampling, and RL, which collectively demonstrate the effectiveness of our proposed method.
>
> We appreciate your suggestion that the main contributions could be made more explicit and easily identifiable. We promise to revise the introduction in the revision to include a clear bullet-point summary of our key contributions at the end of the section.
>
> > *W2: It is recommended that the paper include brief pseudocode for algorithms such as SGRO.*
>
> Thank you for your valuable suggestion! We promise to add a concise Python-style pseudocode illustration of our RL training procedure with SGRO in the revised version, as shown below:
>
> **Input**: Pre-trained DLM $π\_θ$; Group Size $w$; Discount $\lambda$; Learning Rate $\eta$; Response Length $L$
> **Output**: Fine-tuned DLM $π\_{θ^*}$
>
> ```Python
> for each prompt p_0 in dataset:
>     # 1. Roll-out full denoising trajectory
>     r_T = FULL_MASK(L)
>     states, actions = [], []
>     for t = T downto 1:
>         r_t_1 ~ pi_theta(·|p_0, r_t)
>         states.append((p_0, r_t, t))   # s_t
>         actions.append(r_t_1)        # a_t
>
>     # 2. SGRO: compute rewards and returns per group
>     grad = 0.0
>     for g in range(0, T, w):
>         start, end = g, min(g + w, T)
>         # 2-a Intra-sequence rewards inside this group
>         R_intra = 0.0
>         for k in range(start, end):
>             R_intra += TokenVerificationReward(p_0, actions[k])
>             R_intra += PerplexityReward(actions[k])
>         # 2-b Inter-sequence reward (only if the group ends at t=0)
>         R_q = 0.0
>         if end == T:  # covers final step t=0
>             R_q = TaskAccuracyAndFormatReward(actions[end-1])
>         # 2-c Potential-based shaping
>         potential_start = potential_function(states[start])
>         potential_end   = potential_function(states[end-1])
>         shaping = _lambda * potential_end - potential_start
>
>         # group return
>         R_group = R_intra + R_q + shaping
>
>         # 2-d REINFORCE update for steps in this group
>         for k in range(start, end):
>             log_prob = log pi_theta(actions[k] | states[k])
>             grad += ∇_θ log_prob * R_group
>
>     # 3. Parameter update
>     fine_tuned_theta = _theta + _eta * grad
> ```
>
> > *W3&Q4: There are some typos in this paper.*
>
> We appreciate your comment and have carefully proofread the manuscript to correct all typos in the revision.
>
> > *W4&Q1: The rationale for setting the upper bound of the perplexity reward to 100 is unclear. Have your test alternative parameters, and if so, how do they impact the RL training process?*
>
> | Upper Bound      | MATH500  | GPQA     |
> |:---------------- |:--------:|:--------:|
> | LLaDA (Baseline) | 34.4     | 30.3     |
> | LLaDA-MRO-RL-50  | 35.6     | 31.3     |
> | LLaDA-MRO-RL-80  | 37.0     | 33.3     |
> | LLaDA-MRO-RL-100 | **37.4** | **33.8** |
> | LLaDA-MRO-RL-130 | 36.6     | **33.8** |
> | LLaDA-MRO-RL-150 | 35.2     | 32.3     |
>
> We have conducted an ablation study by varying the upper bound of the perplexity reward over a wide range [50,80,100,130,150] and observed the following:
>
> - Performance remains relatively stable within the range of 80 to 130, indicating the robustness of our method and the ease of tuning this hyperparameter.
> - A lower bound may suppress meaningful differences in reward, while a higher bound can introduce excessive variance, both of which can negatively affect training stability.
>
> We promise to add this discussion to the revision.
>
> > *Q2: In Equation 6, what is the rationale for considering only $r_0$ in the inter-sequence correlation rewards? In a Markov setting, this reward should be able to measure the correlation between consecutive state pairs (e.g., $s_t$ and $s_{t-1}$), and all consecutive state pairs should be considered in a complete reasoning trajectory.*
>
> Thanks for your insightful comment! **We would like to clarify that the inter-sequence correlation reward is designed not to measure similarity between sequences at different denoising steps, but rather to assess whether sequences generated under different denoising steps can collaborate to produce a high-quality final rationale**. To capture this cooperative effect, we adopt a **delayed reward** formulation, using only the final reward $r_0$ as a proxy for overall reasoning quality. This approach is inspired by established practices in reinforcement learning literature, where delayed rewards are commonly used to encourage an agent to explore effective an action sequence that lead to a desirable long-term outcome. In our case, the use of $r_0$ allows us to encourage early-stage actions that contribute positively to the final outcome, without being distracted by intermediate, potentially noisy rewards.
>
> > *Q3: In Equation 8, each separate reward has a different range, resulting in varying weights in the total reward. Why is reward normalization not utilized to address this problem?*
>
> Thanks for your insightful comment! We provide both theoretical analysis and empirical evidence showing that SGRO is more effective than reward normalization in reducing reward variance within our MRO.
>
> 1. Our analysis in Appendix B establishes two key results that jointly explain why SGRO is more effective than per-dimension reward normalization in mitigating reward variance.
>
>    - **Variance Amplification from Potential-Based Shaping (Property 1)**
>        Lemma 1 proves that under potential-based reward shaping
>        $R(s_t,a_t)=\hat R(s_t,a_t)+\lambda\Phi(s_{t+1})-\Phi(s_t)$
>        the expected reward remains unchanged, yet the variance strictly increases:
>        $\text{Var}(R(s_t,a_t))>\text{Var}(\hat R(s_t,a_t))$
>        The increased variance originates from the covariance term
>        $-2\lambda\times \text{Cov}(\Phi(s_{t+1}),\Phi(s_t))$
>        which remains **invariant under any affine transformation of the rewards, including min-max normalization, z-score scaling, or simple rescaling. Therefore, reward normalization alone cannot mitigate this covariance-induced variance**.
>
>    - **Variance Reduction via SGRO (Property 2)**
>        Lemma 2 shows that SGRO mitigates this variance by grouping every $w$ consecutive steps and applying the potential difference only once per group:
>        $R^{(w)}(s_t,a_t)=\sum_{i=0}^{w-1}\hat R(s_{t+i},a_{t+i})+\lambda\Phi(s_{t+w})-\Phi(s_t)$
>        By increasing the temporal gap between evaluations of $\Phi$, SGRO effectively reduces the dominant covariance term:
>        $\bigl|\text{Cov}(\Phi(s_{t+w}),\Phi(s_t))\bigr|<\bigl|\text{Cov}(\Phi(s_{t+1}),\Phi(s_t))\bigr|$
>        leading to a strict reduction in reward variance:
>        $\text{Var}(R^{(w)}(s,a))<\text{Var}(R(s,a))$
>
>      In summary, while reward normalization only rescales individual reward components and leaves the covariance structure unchanged, SGRO directly targets and reduces the source of variance introduced by potential-based shaping.
>
> 2. We have further conducted empirical comparisons between SGRO and reward normalization on the MATH500 and GPQA datasets. **As shown below, SGRO consistently yields superior performance, further validating its effectiveness in variance mitigation**:
>
>    | Model/Length      | MATH500  |          | GPQA     |          |
>    | ----------------- |:-------- |:--------:|:-------- |:--------:|
>    |                   | 256      | 512      | 256      | 512      |
>    | LLaDA             | 33.2     | 34.4     | 29.2     | 30.3     |
>    | LLaDA-MRO w/ Norm | 32.8     | 35.0     | 30.0     | 32.8     |
>    | LLaDA-MRO w/ SGRO | **34.2** | **36.2** | **32.1** | **34.3** |
>
> We promise to add this analysis and the empirical comparison in the revision.
>
> ---
>
> We sincerely thank you for your positive feedback on our paper!
> Thank you once again for your time.
>
> Best,
>
> Authors

---

> ### Comment · Area_Chair_oVdo · 2025-08-05
>
> Dear Reviewer 9WDq,
>
> Please review the authors’ rebuttal at your earliest convenience. If you have further questions, use the discussion forum to engage with the authors, and kindly update your review and score as needed.
>
> Thank you for your time and service.
>
> AC

---

> ### Author Response · Authors · 2025-08-06
> **Any Further Concerns About Our Work**
>
> Dear Reviewer 9WDq,
>
> We sincerely appreciate the time and thoughtful feedback you provided on our paper.
>
> During the rebuttal phase, we have carefully addressed your comments and suggestions, including:
>
> * Providing **new experimental results** on MRO under different upper bounds of the perplexity reward. We showed that performance remains relatively stable within the range of 80 to 130, demonstrating the robustness of our method and the ease of tuning this hyperparameter.
>
> * Adding a **comparison between SGRO and reward normalization** on the MATH500 and GPQA datasets. As shown in our results, SGRO consistently outperforms reward normalization, further validating its effectiveness in mitigating reward variance.
>
> * Offering a **theoretical justification** showing why SGRO is more effective than  reward normalization in variance control.
>
> * Including a **pseudocode implementation of SGRO**, which we promise to incorporate into the revised version for greater clarity and reproducibility.
>
> * Addressing other clarifications as requested.
>
> We hope that these additions and responses have adequately addressed your concerns. If so, we would be sincerely grateful if you would consider revisiting your score based on the updated content.
>
> Thank you once again for your valuable time and constructive feedback.
>
> Best regards,
> The Authors

---

> ### Author Response · Authors · 2025-08-08
>
> Dear Reviewer 9WDq,
>
> As the discussion period draws to a close, we would like to follow up once more to check if there are any remaining concerns or questions we can help clarify. We greatly value your feedback and have made a concerted effort to address each of your comments in detail.
>
> We noticed that the current rating remains negative. If you feel that our responses have adequately addressed your concerns, we would be sincerely grateful if you could consider updating your score. Please feel free to let us know if there is anything that still requires clarification.
>
> Thank you again for your time and thoughtful engagement.
>
> Best regards,
> The Authors

---

> > ### Comment · Reviewer_9WDq · 2025-08-08
> >
> > I appreciate the authors’ detailed responses, which address my concerns. I have accordingly raised my score. I strongly recommend that the authors include the additional experiments and analyses in the final version of the paper.

---

> > > ### Author Response · Authors · 2025-08-09
> > >
> > > Dear Reviewer 9WDq,
> > >
> > > Thank you very much for your active participation in the discussion and for providing valuable feedback that has helped us improve the paper. We truly appreciate your decision to re-evaluate your score.
> > >
> > > As promised, we will incorporate the additional analyses and experiments into the final version. In particular, we will include the new results on MRO under different upper bounds of the perplexity reward, which demonstrate that performance remains stable within the range of 80 to 130, confirming both the robustness of our method and the ease of tuning this hyperparameter.
> > >
> > > We are grateful for your constructive suggestions.
> > >
> > > Best regards,
> > > The Authors

---

### Official Review · Reviewer_3ZHw · 2025-07-03

**Clarity:** 3
**Significance:** 2
**Originality:** 3
**Rating:** 5
**Confidence:** 2

**Summary:**

The paper starts with an empirical analysis that identifies the critical issue of why Diffusion Language Models struggle with reasoning tasks due to their inability to capture dependencies between tokens generated in different denoising steps. This paper observes two critical types of token correlations–intra-sequence (within a step) and inter-sequence (across steps)-as essential for coherent and accurate reasoning. To address this, the authors propose a Multi-Reward Optimization (MRO) framework that: (1) introduces multiple reward signals (e.g., token verification, perplexity, reasoning correctness and format adherence) to guide token generation; (2) models the denoising process as a Markov Decision Process (MDP), enabling reinforcement learning-based optimization; and (3) incorporates Step-wise Group Reward Optimization (SGRO) to reduce reward variance and stabilize training. The paper then conducts experiments across various benchmarks demonstrating that MRO significantly enhances reasoning accuracy, while also reducing decoding time by enabling fewer denoising steps.

**Questions:**

1. Can the authors provide a latency analysis of the proposed method compared with vanilla decoding?
2. Could the authors provide analysis on the intra-/inter- sequence correlations with and without the proposed method?

**Ethical Concerns:**

["NO or VERY MINOR ethics concerns only"]

**Final Justification:**

I have carefully read the author's rebuttal. They have added the experiments I suggested in my review. However, I am not an expert in diffusion language models, and after reading the other reviews, I agree with reviewer Sm7r that the computational cost is high. Since the authors incorporated additional experiments I recommended and solve my concern, I will maintain my original score but lower my confidence to 2.

**Limitations:**

Yes

**Quality:**

3

**Strengths And Weaknesses:**

Strengths:
1. The analysis and the identification of the effect of token correlation are insightful and could lead to potential research in the future.
2. The proposed reward design follows the empirical analysis and resolves reward variance.
3. The proposed method improves the performance consistently across 4 benchmarks and especially improves accuracy on lower denoising steps.
4. Overall, the writing is clear and easy to follow.

Weakness:
1. The paper does not provide any efficiency analysis on the decoding time. The proposed method requires the calculation of multiple rewards and policy gradients, which poses a challenge due to latency issues.
2. Although the paper claims that the method could especially benefit reasoning problems with CoTs, it generally mitigates the decoding problem of DLMs and could benefit from further verification on other tasks.
3. The paper could benefit from adding more baselines that require training on the dataset in order to better verify the proposed method.

---

> ### Author Rebuttal · Authors · 2025-07-31
>
> Dear Reviewer 3ZHw,
>
> We sincerely appreciate your constructive and thoughtful feedback.
>
> We greatly appreciate your recognition that the main claim of our analysis on token correlation can "offer valuable insights and could inspire future research", as well as our reward design, empirical improvements across benchmarks, and the clarity of our writing.
>
> Below, we explain the main points you are concerned about.
>
> ---
> >*W1&S1: The paper does not provide any efficiency analysis on the decoding time. The proposed method requires the calculation of multiple rewards and policy gradients, which poses a challenge due to latency issues.*
>
> Thanks for your helpful feedback! Below, we address the concern regarding decoding efficiency and provide empirical evidence to support our claims. **Our results demonstrate that the proposed MRO method is both practical and computationally efficient, making it suitable for real-world applications**.
>
> 1. **We would like to clarify that in both the rejection sampling and RL fine-tuning settings, multi-reward computation and policy gradient updates are performed only during training**. Once the model parameters are updated, decoding proceeds identically to the standard LLaDA baseline [22], with no additional reward computation or forward passes involved. Thus, the end-to-end decoding latency remains unaffected. This is further supported by the latency comparison presented in the table below, which shows that our rejection sampling and RL setups do not introduce noticeable overhead during decoding. Note that all tests are conducted on eight A800 GPUs with a length of 256 under data parallelism, and each setting is run five times to report a range of latency values.
>
>     |Method|MATH500 |  |GPQA|  |
>     |-|:-|:-:|:-|:-:|
>     ||Decoding Time |Score|Decoding Time|Score|
>     |Vanilla Decoding (LLaDA)| 0.35h~0.39h |33.2 | 0.16h~0.21h |29.2 |
>     |LLaDA-RS + MRO| 0.34h~0.37h |34.2 | 0.14h~0.18h |32.1 |
>     |LLaDA-RL + MRO| 0.36h~0.39h |35.2 | 0.16h~0.20h |33.8 |
>
> 2. To further address the concern regarding computational overhead during test-time scaling, we provide a detailed latency comparison between vanilla decoding (i.e., LLaDA) and our proposed MRO, as shown in the table below. Given that TTS requires additional sampling and thus introduces time overhead, we also include a comparison with a confidence-based reward TTS baseline, which involves the same sampling cost as our MRO but does not require extra reward computation. From the results, we observe that **while reward computation does introduce some overhead, it remains within a reasonable range and is justified by the substantial performance gains it brings**. Compared to the confidence-based reward TTS baseline, we further observe that **when excluding the time cost introduced by sampling, our reward design incurs only minimal additional overhead while delivering substantial performance gains**.
>
>     |Method|MATH500|  |GPQA|  |
>     |-|:-|:-:|:-|:-:|
>     ||Decoding Time |Score|Decoding Time|Score|
>     |Vanilla (LLaDA)| 0.35h~0.39h |33.2 | 0.16h~0.21h |29.2|
>     |LLaDA-TTS + Confidence-based Reward| 0.73h~0.81h |35.2 | 0.27h~0.34h |30.6|
>     |LLaDA-TTS + MRO| 0.84h~0.85h |36.0 | 0.31h~0.44h |34.6|
>
> >*W2: This proposed approach generally mitigates the decoding problem of DLMs and could benefit from further verification on other tasks.*
>
> Thanks for the thoughtful suggestion! We have extended our evaluation to a broader range of tasks, including **general knowledge question answering (MMLU)**, **code generation (HumanEval)**, and **instruction following (AlpacaEval2 and Arena-Hard)**, to verify the effectiveness of our proposed MRO further. The results demonstrate that **our MRO achieves strong robustness and can deliver consistent improvements (albeit to varying degrees) across these diverse tasks**. The results and details are as follows.
>
> |Model|MMLU|HumanEval|AlpacaEval2|Arena-Hard|
> |-|:-:|:-:|:-:|:-:|
> |LLaMA-3-8B-Instruction|68.4|59.8|25.3|22.3
> |Mistral-7B-Instruct|60.1|30.5|14.7|12.6
> |Deepseek-LLM-7b-Chat|48.2|26.2|11.2|10.3
> |Qwen2.5-7B-Instruct|71.9|56.7|27.8|25.2
> |||
> |LLaDA|65.5 |47.6 |16.3 |10.0 |
> |LLaDA-RS + MRO|67.5 | 48.1 | **20.2** | 12.3 |
> |LLaDA-RL + MRO|**68.2** | **50.0** | 19.4 | **15.7**|
>
> 1. On general benchmarks such as MMLU and MMLU-Pro, which span multiple domains including biology and physics, our method continues to demonstrate effectiveness. This suggests that MRO is not limited to mathematical reasoning tasks. Specifically, we replace Countdown and Sudoku in our training set with a subset of the MMLU training set for both rejection sampling and reinforcement learning.
>
> 2. On the code generation task, we train the model using the CodeAlpaca-20K dataset. Our method still yields notable improvements under both RL and rejection sampling settings.
>
> 3. To evaluate instruction-following capabilities, we train the model on the Alpaca dataset. Note that during the computation of inter-sequence rewards for this task, we replace the standard answer-matching metric with a learned reward model, which allows for more nuanced assessment of response quality.
>
>
> >*W3: The paper could benefit from adding more baselines that require training on the dataset in order to verify the proposed method better.*
>
> Thanks for your valuable feedback! In response, we have achieved several additional baselines, all trained on the same dataset (DeepScaleR + Countdown/Sudoku) using identical hyperparameters. Our experimental results show that **LLaDA combined with our proposed MRO consistently achieves the best performance**. Specifically, we designed the following experiments:
>
> 1. **Reward variants within the LLaDA model**: We have achieved baselines using different reward functions for both rejection sampling and reinforcement learning to the LLaDA model, including LLaDA-$R^{tv}\_{t}$, LLaDA-$R^{ppl}\_{t}$, and LLaDA-$R^{q}\_{0}$. _Note that these results are already present in our initial submission and are used as baselines in training, as shown in Figure 6._
>
> 2. **Baselines using other foundation models trained on the dataset**: We have achieved RL fine-tuning baselines using other popular foundation models, such as LLaMA-3-8B-Instruction, Deepseek-LLM-7B-Chat, and Mistral-7B-Instruct, trained on the same dataset using our reward designs and the REINFORCE++ algorithm. For these models, we use a delayed reward at the end of the sampled reasoning trajectory based on the final output, following the same reward evaluation protocol.
>
> 3. **Rejection sampling with other foundation models**: We also perform rejection sampling experiments with other foundation models, where multiple reasoning paths are sampled and the one with the highest reward score is selected using our proposed reward design.
>
>
> The results are summarized below. For consistency, the maximum length is set to 512 for MATH500 and GPQA, and to 128 for Countdown.
>
> |Model||Rejection Sampling|||Reinforcement Learning||
> |-|:-:|:-:|:-:|:-:|:-:|:-:|
> ||MATH500|GPQA|Countdown|MATH500|GPQA|Countdown|
> |_**Training with Other Foundation Models**_|
> |LLaMA-3-8B-Instruction|18.8(+0.4)|26.8(+1.2)|4.5(+4.5)|17.4(-1.0)|28.7(+3.1)|0.4(+0.4)
> |Mistral-7B-Instruct|13.6(+0.5)|24.2(-0.5)|3.2(+3.2)|14.0(+0.9)|23.2(-1.5)|3.4(+3.4)
> |Deepseek-LLM-7B-Chat|11.2(+5.2)|20.7(+1.2)|3.8(+3.8)|10.0(+4.0)|22.0(+2.8)|2.6(+2.6)
> |_**Reward Variants within the LLaDA**_|||
> |LLaDA-$R^{tv}_{t}$|36.0(+1.6)|33.2(+2.9)|**23.4(+12.2)**|36.2(+1.8)|32.7(+2.4)|25.3(+11.2)|
> |LLaDA-$R^{ppl}_{t}$|34.0(-0.4)|30.5(+0.2)|14.1(+2.9)|33.6(-0.8)|30.8(+0.5)|18.9(+4.8)|
> |LLaDA-$R^{q}_{0}$|35.0(+0.6)|32.8(+2.5)|19.8(+8.6)|34.8(+0.4)|31.2(+0.9)|23.5(+9.4)|
> |LLaDA + MRO|**36.2(+1.8)**|**34.3(+4.0)**|22.0(+7.9)|**37.4(+3.0)**|**33.8(+3.5)**|**27.2(+13.1)**|
>
> From the results, we can observe the following:
>
> - When comparing different reward variants, our proposed MRO consistently achieves the best performance. The results confirm the validity of our design approach, which focuses on improving intra-/inter-sequence correlation through carefully crafted rewards. **This further demonstrates that the improvements are not primarily attributable to the dataset itself, but rather stem from our method's ability to address the overlooked issue of correlation in decoder language models.**
>
> - Applying the same-scale RL fine-tuning or rejection sampling on other foundation models (such as LLaMA-3 and Mistral) also leads to performance improvements on this dataset. However, the gains (_as indicated in parentheses in the table_) are generally smaller compared to those achieved by LLaDA. This can be attributed to the fact that these models have already undergone RLHF training and are relatively well-aligned. In contrast, LLaDA has not been exposed to any prior RL training, leaving more room for improvement.
>
>
> >*S2: Could the authors provide analysis on the intra-/inter- sequence correlations with and without the proposed method?*
>
> Thanks for this helpful suggestion! We have conducted a quantitative analysis of intra- and inter-sequence correlations with and without our proposed MRO. The results show that **MRO can significantly enhance both intra- and inter-sequence correlations**. Specifically, we randomly sampled 200 examples from the MATH500 and GPQA datasets, and computed the differences in intra- and inter-sequence reward scores across five decoding runs with different seeds. The detailed comparison is presented below.
>
> | Model | MATH500 | | GPQA||
> | - | :- | :-: | :- | :-: |
> |   | intra-corr | inter-corr | intra-corr | inter-corr |
> | LLaDA | 3.44±0.18  |  1.02±0.14 | 2.76±0.21  |  1.02±0.15 |
> | LLaDA-RS-MRO | 3.79±0.16 | **1.58**±0.12 | 3.34±0.19  |  1.27±0.13 |
> | LLaDA-RL-MRO | **4.12**±0.24 | 1.42±0.11 | **3.55**±0.17 | **1.43**±0.12 |
>
> We promise to add these related discussions in the revised version.
>
> ---
> We sincerely thank you for your positive feedback on our paper!
>
> Best,
>
> Authors

---

> > ### Comment · Reviewer_3ZHw · 2025-08-08
> >
> > Thanks for the response. I decide to keep my score.

---

> > > ### Author Response · Authors · 2025-08-09
> > >
> > > Dear Reviewer,
> > >
> > > Thank you for your valuable suggestions and feedback. Your comments have been very helpful in identifying areas for improvement.
> > >
> > > Best regards,
> > > The Authors

---

### Decision · Program_Chairs · 2025-09-17

**Decision:**

Accept (poster)

**Comment:**

This paper introduces MRO, a multi-reward optimization framework to address token-correlation deficits in diffusion language models. The method is validated across diverse reasoning and general tasks, showing consistent gains and meaningful ablations. Reviewers raise concerns about mathematical precision, clarity of reward taxonomy, limited theory, and scalability, but these are presentation issues rather than fatal flaws. With camera-ready fixes (notation, compute reporting, clearer reward framing), the work makes a timely and solid contribution.